# Provable Quantum Algorithm Advantage for Gaussian Process Quadrature

**Cristian A. Galvis-Florez**                                    *cristian.galvis@aalto.fi*
*Department of Electrical Engineering and Automation*
*Aalto University, Finland*

**Ahmad Farooq**                                    *ahmad.farooq@aalto.fi*
*Department of Electrical Engineering and Automation*
*Aalto University, Finland*

**Simo Särkkä**                                    *simo.sarkka@aalto.fi*
*Department of Electrical Engineering and Automation*
*Aalto University, Finland*

**Reviewed on OpenReview:** *https: // openreview. net/ forum? id=K6CvWPtF62*

## Abstract

The aim of this paper is to develop novel quantum algorithms for Gaussian process quadrature methods. Gaussian process quadratures are numerical integration methods where Gaussian processes are used as functional priors for the integrands to capture the uncertainty arising from the sparse function evaluations. Quantum computers have emerged as potential replacements for classical computers, offering exponential reductions in the computational complexity of machine learning tasks. In this paper, we combine Gaussian process quadratures and quantum computing by proposing a quantum low-rank Gaussian process quadrature method based on a Hilbert space approximation of the Gaussian process kernel and enhancing the quadrature using a quantum circuit. The method combines the quantum phase estimation algorithm with the quantum principal component analysis technique to extract information up to a desired rank. Then, Hadamard and SWAP tests are implemented to find the expected value and variance that determines the quadrature. We use numerical simulations of a quantum computer to demonstrate the effectiveness of the method. Furthermore, we provide a theoretical complexity analysis that shows a polynomial advantage over classical Gaussian process quadrature methods. The code is available at https://github.com/cagalvisf/Quantum_HSGPQ.

# 1 Notation

### Classical notation

| | |
|---|---|
| $a \sim P$ | Random variable a has distribution $P$ |
| $\mathcal{D}$ | Data set $\mathcal{D} = \{(\mathbf{x}_i, \mathbf{y}_i) \mid i = 1, \ldots, N\}$ |
| $E[f(\mathbf{x}_*) \mid \mathcal{D}]$ | Expectation of $f(\mathbf{x})$ at $\mathbf{x} = \mathbf{x}_*$ given the data set $\mathcal{D}$ |
| $k(\mathbf{x}, \mathbf{x}')$ | Kernel function evaluated at $\mathbf{x}$ and $\mathbf{x}'$ |
| $K$ or $K(X, X)$ | $N \times N$ covariance (or Gram) matrix |
| $\mathcal{N}(\boldsymbol{x} \mid \boldsymbol{\mu}, \boldsymbol{\Sigma})$ | Gaussian distribution over $\boldsymbol{x}$ with mean $\boldsymbol{\mu}$ and covariance $\boldsymbol{\Sigma}$ |
| $V[f(\mathbf{x}_*) \mid \mathcal{D}]$ | Variance of $f(\mathbf{x})$ at $\mathbf{x} = \mathbf{x}_*$ given the data set $\mathcal{D}$ |
| $\mathbf{x}^\top$ | Transpose of the vector $\mathbf{x}$ |
| $\delta_{ij}$ | Kronecker delta, $\delta_{ij} = 1$ iff $i = j$ and 0 otherwise |
| $\lambda$ | Eigenvalue of an eigenfunction or singular value of a matrix |
| $\sigma$ | Standard deviation of noise or a scale parameter |
| $\phi_i(\mathbf{x})$ | The $i$th eigenfunction of the Laplace operator |

### Quantum computing notation

| | |
|---|---|
| $\otimes$ | Tensor product |
| $\lvert \cdot \rangle$ | Ket notation of a column vector |
| $\langle \cdot \rvert$ | Bra notation for the conjugate transpose of $\lvert \cdot \rangle$ |
| $\langle \cdot \vert \cdot \rangle$ | Braket or inner product between two vectors |
| $\lvert \cdot \rangle \lvert \cdot \rangle$ | Tensor product of the two separate ket vectors |
| $\lvert \cdot \rangle \langle \cdot \rvert$ | Outer product between two vectors |
| $\lvert j \rangle$ | Orthonormal basis state of integer $j$ |
| $\lvert j_1 \rangle \otimes \lvert j_2 \rangle =$ | Tensor product of quantum systems |
| $\lvert j_1 \rangle \lvert j_2 \rangle = \lvert j_1 j_2 \rangle$ | |
| $\lvert \xi \rangle$ | Used for intermediate states in the quantum algorithms |
| $\langle \psi \rvert U \lvert \psi \rangle$ or $\langle U \rangle_\psi$ | Expected value of U over the quantum state $\langle \psi \rvert$ |
| $\lvert \psi_{\mathbf{X}} \rangle$ | Quantum state encoding the data matrix $\mathbf{X}$ |
| $\text{Im}(\cdot)$ | Imaginary part of the value |
| $p_i$ | Probability to measure the quantum system in the state $\lvert i \rangle$ |
| $\text{Re}(\cdot)$ | Real part of the value |
| $\text{tr}_n(\cdot)$ | Partial trace over $n$th subsystem |
| $U$ | Unitary quantum operator. Matrix with components $U_{ij} = \langle i \rvert U \lvert j \rangle$ |
| $\tilde{\sigma}$ | Density matrix |
| $\rho$ | Density matrix |
| $\tilde{\rho}^{(n)}$ | Density matrix of $n$th subsystem |

## 2 Introduction

The integration of analytically intractable functions is usually addressed by numerical methods such as classical quadrature rules (Davis & Rabinowitz, 1984). However, alternatives to the classical quadrature rules are Bayesian quadratures which formulate the numerical integration problem as a Bayesian inference problem over the integral value, hence allowing for quantification of uncertainty arising from the finite number of function evaluations (O'Hagan, 1991; Minka, 2000; Hennig et al., 2022). In particular, a common way is to consider a Gaussian prior distribution for the integrand, which leads to so-called Gaussian process quadratures (GPQs) (Minka, 2000; Hennig et al., 2022). However, the GPQ methods struggle with computational efficiency, especially when handling large datasets since the algorithm complexity scales as $O(N^3)$, with $N$ being the number of evaluation points, which is a problem inherited from the similar complexity challenge in Gaussian process regression.

Quantum computers have emerged as a novel approach for general computational purposes, showing even exponential speedups in computational complexity over their classical counterparts. Quantum computers exploit superposition, entanglement, and interference to perform a task, which substantially reduces the computational complexity of an algorithm (Nielsen & Chuang, 2011). The Harrow-Hassidim-Lloyd (HHL) algorithm (Harrow et al., 2009), quantum kernel methods (Schuld & Killoran, 2019), quantum phase estimation (QPE, Kitaev, 1995), and quantum amplitude amplification (QAA, Brassard et al., 2002) are quantum algorithms that offer a computational advantage in different numerical tasks (Schuld et al., 2016; Shor, 1994; Grover, 1996; Wiedemann et al., 2023; Canatar et al., 2023). Quantum algorithms for numerical integration in the classical setting have been proposed by Yu et al. (2020) and Shu et al. (2024). Furthermore, Carrera Vazquez & Woerner (2021) and Martínez de Lejarza et al. (2023) implement quantum algorithms that have a quadratic speedup with respect to Monte Carlo integration methods. These integration methods for classical numerical integration on a quantum computer have been proposed as an application of efficient state preparation for quantum amplitude estimation (Martínez de Lejarza et al., 2023).

The contribution of this paper is to propose a quantum Bayesian quadrature algorithm for numerical integration. A naive numerical integration algorithm using a Gaussian quadrature on a non-quantum classical computer would have a complexity of $O(N^3)$, which is inherited from the complexity of Gaussian process regression (Rasmussen et al., 2006). In this paper, we develop a low-rank quantum Gaussian process quadrature method for numerical integration employing a low-rank Gaussian process prior, using the GPR method in Farooq et al. (2024) as the starting point. Our algorithm harnesses the quantum Fourier transform, quantum phase estimation, and quantum principal component analysis to exploit the superposition of quantum states for computing the matrix inverse in GPQ, thereby providing an advantage over the classical approach. The proposed method begins by implementing a Hilbert space kernel approximation (Solin & Särkkä, 2020) using a classical computer to evaluate the basis function integrals at the evaluation points. Then, the resulting data matrix needs to be loaded into a quantum computer efficiently, and for this purpose, the approximate quantum amplitude encoding scheme is implemented (Nakaji et al., 2022). Using quantum principal component analysis (qPCA) (Lloyd et al., 2014) and QPE we extract information about the dominant eigenvalues of the previously encoded data matrix. This information is used to implement conditional rotations that allow the estimation of the expected value and the variance of the quadrature using a Hadamard and SWAP test respectively (Schuld & Petruccione, 2021; Buhrman et al., 2001). We provide the numerical simulation of our proposed solution for quantum low-rank Gaussian process quadrature in a classical processor that simulates a quantum computer, allowing us to compare it with classical Gaussian process quadrature methods.

Our method also provides a polynomial computational advantage compared to classical methods for Gaussian process quadrature. The quantum algorithm implemented in this paper delivers a complexity $O(\log(M)\kappa^2 s^2/\epsilon)$, where $\kappa$ is the condition number, $s$ is the sparsity of the matrix, $\epsilon$ is the desired level of accuracy, and $M$ is the number of basis functions used in the Hilbert space approximation (usually $M \ll N$), with given quantum data, and $O(\text{poly}(\log(NM))\log(M)\kappa^2\epsilon^{-3})$ when data is given classically. The computational complexity of a GPQ method implemented on a classical computer with the Hilbert space method would be $O(M^3)$ and hence, the advantage is polynomial.

The paper is organized as follows. In Section 3 we briefly review the Gaussian process quadrature methods and show how to approximate them with the Hilbert space method. In Section 4 we summarize the generic

quantum algorithms used in this paper. Section 5 encompasses the core of this work, presenting the quantum low-rank Gaussian process quadrature method and discussing its computational complexity. In Section 6 numerical simulations using a classical computer are shown. Section 7 concludes the work.

## 3 Hilbert Space Approximation for Gaussian Process Quadrature

Gaussian processes $f \sim \mathcal{GP}(0, k(\mathbf{x}, \mathbf{x}'))$ can be used as functional prior distributions to model unknown functions $f$ over a $d$-dimensional inputs $\mathbf{x}$ on to a space $\Omega \subseteq \mathbb{R}^d$. The covariance function $k(\mathbf{x}, \mathbf{x}')$ defines the properties of the unknown function $f$ and the mean can often be assumed to be zero, as we do here. Regression of a function $f(\mathbf{x})$ can be performed using the Gaussian process as the prior distribution in a process known as Gaussian process regression (GPR) (Rasmussen et al., 2006). Then, a set of $N$ noisy measurements $\mathcal{D} = \{\mathcal{X} = \{\mathbf{x}_1, \cdots, \mathbf{x}_N\}, \mathbf{y} = (y_1, \cdots, y_N)\}$ with $y_i = f(\mathbf{x}_i) + \varepsilon_i$, where $\varepsilon_i \sim \mathcal{N}(0, \sigma^2)$, induce a posterior Gaussian distribution $p(f(\mathbf{x}_*) \mid \mathbf{x}_*, \mathcal{D}) = \mathcal{N}(f(\mathbf{x}_*) \mid \mathrm{E}[f(\mathbf{x}_*)], \mathrm{V}[f(\mathbf{x}_*)])$ on a new point $\mathbf{x}_*$ given by

$$\mathrm{E}[f(\mathbf{x}_*) \mid \mathcal{D}] = \mathbf{k}_*^\top (\mathbf{K} + \sigma^2 \mathbf{I})^{-1} \mathbf{y}, \tag{1}$$

$$\mathrm{V}[f(\mathbf{x}_*) \mid \mathcal{D}] = k(\mathbf{x}_*, \mathbf{x}_*) - \mathbf{k}_*^\top (\mathbf{K} + \sigma^2 \mathbf{I})^{-1} \mathbf{k}_*. \tag{2}$$

Here, $\mathbf{K}$ is a $N \times N$ matrix with entries $K_{ij} = k(\mathbf{x}_i, \mathbf{x}_j)$ and $\mathbf{k}_*$ is a vector with the $i$th entry being $k(\mathbf{x}_*, \mathbf{x}_i)$.

The choice of the covariance or kernel function defines the properties of the Gaussian process regression solution. A common kernel choice for GPR is the squared exponential covariance function (Rasmussen et al., 2006)

$$k(\mathbf{x}, \mathbf{x}') = \sigma_f^2 \exp\left(-\frac{1}{2l^2} \|\mathbf{x} - \mathbf{x}'\|^2\right), \tag{3}$$

where $\sigma_f$ and $l$ are the signal scale and length scale hyperparameters respectively. Gaussian processes can also be used to construct Bayesian quadrature rules leading to so-called Gaussian process quadratures (Minka, 2000; Hennig et al., 2022) which we discuss next.

The objective of a Gaussian process quadrature is to estimate the integral of a given function $f(\mathbf{x})$ over a domain $\mathbf{x} \in \Omega$ against a weight function $\mu(\mathbf{x})$, that is,

$$\mathcal{I} = \int_\Omega f(\mathbf{x})\mu(\mathbf{x})d\mathbf{x}. \tag{4}$$

To perform this estimation, we consider a Gaussian process approximation of the function $f$ conditional on a finite set of evaluation points $\mathbf{y}$ with $y_i = f(\mathbf{x}_i) + \epsilon_i$ at some given evaluation points $\mathbf{x}_i$. Then the conditional expected value of the integral given this data $\mathcal{D}$ is

$$\hat{\mathcal{I}} = \int_\Omega \mathrm{E}[f(\mathbf{x}) \mid \mathcal{D}]\mu(\mathbf{x})d\mathbf{x}$$
$$= \left[\int_\Omega k(\mathcal{X}, \mathbf{x})\mu(\mathbf{x})d\mathbf{x}\right]^\top (\mathbf{K} + \sigma^2 I)^{-1}\mathbf{y}, \tag{5}$$

which can be used as an estimator for the integral $\hat{\mathcal{I}} \approx \mathcal{I}$. Similarly, we can evaluate the variance of the estimator. The resulting conditional expected value and variance are given by (Hennig et al., 2022; Karvonen & Särkkä, 2017)

$$Q_{\mathrm{BQ}} := k_\mu(\mathcal{X})^\top (\mathbf{K} + \sigma^2 \mathbf{I})^{-1} \mathbf{y}, \tag{6}$$

$$V_{\mathrm{BQ}} := \mu(k_\mu) - k_\mu(\mathcal{X})^\top (\mathbf{K} + \sigma^2 I)^{-1} k_\mu(\mathcal{X}), \tag{7}$$

where the $i$-th component of the vector $k_\mu(\mathcal{X})$ is $k_\mu(\mathbf{x}_i) = \int_\Omega k(\mathbf{x}_i, \mathbf{x})\mu(\mathbf{x})d\mathbf{x}$ and the scalar $\mu(k_\mu) = \int_\Omega \int_\Omega k(\mathbf{x}, \mathbf{x}')\mu(\mathbf{x})d\mathbf{x}\mu(\mathbf{x}')d\mathbf{x}'$. The resulting quadrature $Q_{\mathrm{BQ}}$ approximates the integral $\mathcal{I}$ by performing integration over the kernel rather than the function $f$ itself, and by using these values it forms weights for

the function values. There exist a number of methods that can be used to reduce the complexity of Gaussian process regression such as inducing point methods (Quinonero-Candela & Rasmussen, 2005) and Hilbert space kernel approximations (Solin & Särkkä, 2020), which can be used to speed up the GPQ rule.

In this paper, we implement the Hilbert space method of Solin & Särkkä (2020) on the quadrature rule to reduce the complexity of the operation. We start by considering the eigenvalue problem of the Laplace operator in a well-behaved domain $\zeta$ such that the eigenfunctions and eigenvalues of the operator exist. The solution would give a set of eigenfunctions $\phi_j(\mathbf{x}) \in \zeta$ with correspondent real and positive eigenvalues $\lambda_j$ (Arfken et al., 2005). The eigenfunction are orthonormal respect to the inner product $\int \phi_j \phi_i d\mathbf{x} = \delta_{ij}$, defining a Hilbert space over $\zeta$.

If the kernel function is isotropic, that is, $k(\mathbf{x}, \mathbf{x}') = k(||\mathbf{x} - \mathbf{x}'||)$, then the covariance function can be approximated in the domain $\zeta$ as (Solin & Särkkä, 2020)

$$k\left(\mathbf{x}, \mathbf{x}'\right) \approx \sum_{j=1}^{M} S\left(\sqrt{\lambda_j}\right) \phi_j(\mathbf{x}) \phi_j\left(\mathbf{x}'\right), \tag{8}$$

where the spectral density $S(\omega)$ is the Fourier transform $(\mathcal{F})$ of the scalar covariance function $k(\tau) \xrightarrow{\mathcal{F}} S(\omega)$ where $\tau = ||\mathbf{x} - \mathbf{x}'||$.

Equations (6) and (7) can be written in terms of the approximated kernel, giving

$$Q_{\text{BQ}} \approx \mathbf{\Phi}_\mu^\top (\mathbf{\Phi}^\top \mathbf{\Phi} + \sigma^2 \mathbf{\Lambda}^{-1})^{-1} \mathbf{\Phi}^\top \mathbf{y}, \tag{9}$$

$$V_{\text{BQ}} \approx \sigma^2 \mathbf{\Phi}_\mu^\top (\mathbf{\Phi}^\top \mathbf{\Phi} + \sigma^2 \mathbf{\Lambda}^{-1})^{-1} \mathbf{\Phi}_\mu, \tag{10}$$

where the vector $\mathbf{\Phi}_\mu$ has the components $\Phi_{\mu_i} = \int_\Omega \phi_i(\mathbf{x}) \mu(\mathbf{x}) d\mathbf{x}$, $\mathbf{\Lambda}$ is a diagonal matrix with components $\mathbf{\Lambda}_{jj} = S(\sqrt{\lambda_j})$ and the matrix $\mathbf{\Phi}$ has components $\Phi_{ij} = \phi_j(\mathbf{x}_i)$. The approximation of the kernel now depends on the domain $\zeta$, which is not necessarily the same as the integration domain $\Omega$. The matrix in Eqs. (9) and (10) that needs to be inverted now has rank $M$ instead of $N$. This kernel approximation reduces the computational complexity involved in matrix inversion provided that $M \ll N$.

The expected value and variance expressions of the quadrature can be decomposed using singular value decomposition (SVD) to embed them into quantum states (Schuld et al., 2016). Some modifications of Eqs. (9) and (10) must be done before the implementation of the algorithm (Farooq et al., 2024). For the GPQ, we need to obtain the eigenvalues and eigenvectors of $\left(\mathbf{\Phi}^\top \mathbf{\Phi} + \sigma^2 \mathbf{\Lambda}^{-1}\right)$, which we aim to express in terms of $\mathbf{\Phi}^\top \mathbf{\Phi}$. By writing the expected value and variance of the quadrature in terms of this common set of eigenvectors, the expected value and variance can be written in terms of quantum states.

Similarly to Farooq et al. (2024) we start by considering $\mathbf{X} = \mathbf{\Phi}\sqrt{\mathbf{\Lambda}} \in \mathbb{R}^{N \times M}$, where $\sqrt{\mathbf{\Lambda}}$ is a diagonal matrix with elements $\sqrt{\mathbf{\Lambda}_{ii}} = \sqrt{S(\sqrt{\lambda_i})}$, the quadrature equations take the form

$$Q_{\text{BQ}} = \mathbf{X}_\mu^\top \left(\mathbf{X}^\top \mathbf{X} + \sigma^2 \mathbf{I}\right)^{-1} \mathbf{X}^\top \mathbf{y}, \tag{11}$$

$$V_{\text{BQ}} = \sigma^2 \mathbf{X}_\mu^\top \left(\mathbf{X}^\top \mathbf{X} + \sigma^2 \mathbf{I}\right)^{-1} \mathbf{X}_\mu, \tag{12}$$

where $\mathbf{X}_\mu^\top = \mathbf{\Phi}_\mu^\top \sqrt{\mathbf{\Lambda}}$. Now the eigenvectors of $\mathbf{X}^\top \mathbf{X} + \sigma^2 \mathbf{I}$ are the same as those of $\mathbf{X}^\top \mathbf{X}$. Previous equations are referred to as the Hilbert space quadrature (HSQ). In this paper, we use a quantum computer to calculate these quantities, which we call the quantum Hilbert space quadrature method (QHSQ). The variables $\mathbf{X}$ and $\mathbf{X}_\mu$ are computed on a classical computer before we feed them into the quantum computer.

Using singular value decomposition (SVD) to the data matrix $\mathbf{X}$ it is possible to write the expected value and variance of the GPQ in terms of quantum states. Let $\mathbf{X} = \mathbf{U}\mathbf{\Sigma}\mathbf{V}^\top$ be the SVD of the data matrix $\mathbf{X}$. The matrix $\mathbf{\Sigma} \in \mathbb{R}^{R \times R}$ is diagonal, containing the real singular values $\lambda_1, \lambda_2, \ldots, \lambda_R$, being $R$ the rank of the matrix $\mathbf{X}$. The orthogonal matrices $\mathbf{U} \in \mathbb{R}^{N \times R}$ and $\mathbf{V} \in \mathbb{R}^{R \times M}$ correspond to the left and right singular vectors, respectively. Thereof, the SVD of $\mathbf{X}^\top \mathbf{X} + \sigma^2 \mathbf{I} = \mathbf{V}\mathbf{\Sigma}'\mathbf{V}^\top$ leads to the diagonal matrix $\mathbf{\Sigma}'$ with elements $\mathbf{\Sigma}'_{ii} = \lambda_i^2 + \sigma^2$. Then, the eigendecomposition of $\left(\mathbf{X}^\top \mathbf{X} + \sigma^2 \mathbf{I}\right)^{-1} \mathbf{X}^\top$ is given by

$$\left(\mathbf{X}^\top \mathbf{X} + \sigma^2 \mathbf{I}\right)^{-1} \mathbf{X}^\top = \mathbf{V}\mathbf{\Sigma}''\mathbf{U}^\top = \sum_{r=1}^{R} \frac{\lambda_r}{\lambda_r^2 + \sigma^2} \mathbf{v}_r \mathbf{u}_r^\top, \tag{13}$$

where $\mathbf{\Sigma}''$ is diagonal with components $\mathbf{\Sigma}''_{ii} = \frac{\lambda_i}{\lambda_i^2+\sigma^2}$ and the vectors $\mathbf{u}_r$ and $\mathbf{v}_r$ correspond to the $r$-th column vectors of the matrices $\mathbf{U}$ and $\mathbf{V}$ respectively. Then, using the SVD, the expected value and variance of the GPQ can be written as

$$Q_{\mathrm{BQ}} = \sum_{r=1}^{R} \frac{\lambda_r}{\lambda_r^2+\sigma^2} \mathbf{X}_\mu^\top \mathbf{v}_r \mathbf{u}_r^\top \mathbf{y}, \tag{14}$$

$$V_{\mathrm{BQ}} = \sigma^2 \sum_{r=1}^{R} \frac{1}{\lambda_r^2+\sigma^2} \mathbf{X}_\mu^\top \mathbf{v}_r \mathbf{v}_r^\top \mathbf{X}_\mu. \tag{15}$$

The SVD in the expected value and variance of the GPQ allows us to write them as expected values of quantum operators, extending the complexity reduction provided by the Hilbert space approximation of the kernel into the quantum algorithm.

## 4 Quantum Algorithm Background

In this section, we will review the quantum algorithms needed to implement the quantum low-rank Gaussian process quadrature. Readers who are familiar with fundamental quantum algorithms can skip this section and directly move to the next Section 5. We have provided more details on the representation of quantum states, unitary transformation operations, and quantum measurements in Appendices A.1, A.2, and A.3.

### 4.1 Quantum Fourier transform

The quantum Fourier transform (QFT) performs the same transformation as the discrete Fourier transform. It was first used in Shor's integer factoring algorithm (Shor, 1994). Nowadays, QFT is an essential part of various quantum linear algebra algorithms such as quantum phase estimation, matrix inversion problems, and qPCA (Nielsen & Chuang, 2011; Harrow et al., 2009; Lloyd et al., 2014). The discrete Fourier transform takes an input vector $\mathbf{x}$ and transforms it into another vector in the frequency domain $\mathbf{y}$ through

$$y_k = \frac{1}{\sqrt{2^n}} \sum_{j=0}^{2^n-1} e^{\frac{2\pi i j k}{2^n}} x_j, \qquad k = 0, \cdots, 2^n - 1. \tag{16}$$

Similarly, QFT is defined as a linear operator that acts on an orthonormal basis $|0\rangle, \ldots, |N-1\rangle$ where $N = 2^n$. The QFT acts over the basis element $|j\rangle$ as

$$|j\rangle \rightarrow \frac{1}{\sqrt{2^n}} \sum_{k=0}^{2^n-1} e^{\frac{2\pi i j k}{2^n}} |k\rangle. \tag{17}$$

We can express this as the unitary transformation

$$U = \frac{1}{\sqrt{2^n}} \sum_{k=0}^{2^n-1} \sum_{j=0}^{2^n-1} e^{\frac{2\pi i j k}{2^n}} |k\rangle \langle j|, \tag{18}$$

acting on an arbitrary state $|\psi\rangle$, which can be expressed as

$$|\psi\rangle = \sum_{j=0}^{2^n-1} x_j |j\rangle \rightarrow \sum_{k=0}^{2^n-1} y_k |k\rangle. \tag{19}$$

To understand how the QFT has been implemented on quantum computers, it is helpful to express the state $|j\rangle$ in a binary representation as $j = j_1 j_2 \cdots j_n$, where $j_1, j_2, \cdots, j_n$ are either 0 or 1. For example, we can represent the state $|j\rangle$ when $j = 2$ as a unit vector using binary representation: $|2\rangle = |10\rangle = |1\rangle \otimes |0\rangle = \begin{pmatrix} 0 & 0 & 1 & 0 \end{pmatrix}^\top$, where single qubit states are $|0\rangle = \begin{pmatrix} 1 \\ 0 \end{pmatrix}$ and $|1\rangle = \begin{pmatrix} 0 \\ 1 \end{pmatrix}$. The binary representation of an

entire number can be written as $j = j_1 2^{n-1} + j_2 2^{n-2} + \cdots + j_n 2^0$. Similarly, a decimal number $0.j_1 j_2 \cdots j_l$ can be expressed in a binary representation as $j_1/2 + j_2/4 + \cdots + j_l/2^l$. The QFT in Eq. (17) can be written in product form following Nielsen & Chuang (2011) as

$$|j_1, \cdots, j_n\rangle \to \frac{1}{2^{n/2}} \left[ \left( |0\rangle + e^{2\pi i 0.j_n} |1\rangle \right) \left( |0\rangle + e^{2\pi i 0.j_{n-1} j_n} |1\rangle \right) \cdots \left( |0\rangle + e^{2\pi i 0.j_1 j_2 \cdots j_n} |1\rangle \right) \right]. \quad (20)$$

This expression can be implemented using a quantum circuit composed only of the Hadamard gate $H = \frac{1}{\sqrt{2}} \begin{bmatrix} 1 & 1 \\ 1 & -1 \end{bmatrix}$, and the two-qubit controlled rotation $CR_k = \begin{bmatrix} I & 0 \\ 0 & R_k \end{bmatrix}$ given in a block-diagonal form, where $R_k = \begin{bmatrix} 1 & 0 \\ 0 & e^{2\pi i/2^k} \end{bmatrix}$. The action of the Hadamard gate on the target quantum state $|j_t\rangle$ is

$$H |j_t\rangle = \frac{1}{\sqrt{2}} \left( |0\rangle + e^{\frac{2\pi i}{2} j_t} |1\rangle \right) = \frac{1}{\sqrt{2}} \left( |0\rangle + e^{2\pi i 0.j_t} |1\rangle \right), \quad (21)$$

where we represented the $j_t/2$ as a decimal number $0.j_t$. The Eq. (21) can be seen as follows: when $j_t = 0$, we have $H |0\rangle = \frac{1}{\sqrt{2}} (|0\rangle + |1\rangle)$, and when $j_t = 1$ we have $H |1\rangle = \frac{1}{\sqrt{2}} (|0\rangle - |1\rangle)$. On the other hand, the operation of $CR_k$ on a two-qubit state $|j_c j_t\rangle$, where the first qubit is the control and the second is the target, does not affect the second qubit if the first is in the zero state $CR_k |0 j_t\rangle = |0 j_t\rangle$, and when the control qubit in one state is $CR_k |1 j_t\rangle = e^{\frac{2\pi i}{2^k} j_t} |1 j_t\rangle$.

We start with an $n$-qubit input quantum state represented in $|j_1 \cdots j_n\rangle$. Applying the Hadamard gate to the first qubit gives the following quantum state

$$H |j_1\rangle |j_2 \cdots j_n\rangle = \frac{1}{2^{1/2}} \left( |0\rangle + e^{\frac{2\pi i}{2} j_1} |1\rangle \right) |j_2 \cdots j_n\rangle = \frac{1}{2^{1/2}} \left( |0\rangle + e^{2\pi i 0.j_1} |1\rangle \right) |j_2 \cdots j_n\rangle. \quad (22)$$

Then, the operation of the controlled-$R_2$ gate to the first two qubits results in

$$CR_2 |j_1 j_2\rangle |j_3 \cdots j_n\rangle = \frac{1}{2^{1/2}} \left( |0\rangle + e^{\frac{2\pi i}{2} j_1} e^{\frac{2\pi i}{4} j_2} |1\rangle \right) |j_2 \cdots j_n\rangle = \frac{1}{2^{1/2}} \left( |0\rangle + e^{2\pi i 0.j_1 j_2} |1\rangle \right) |j_2 \cdots j_n\rangle. \quad (23)$$

Similarly, applying the controlled gates $R_k$ for $k = 3, \ldots, n$ on the first qubit yields the following expression:

$$|\xi_1\rangle = \frac{1}{2^{1/2}} \left( |0\rangle + e^{2\pi i (j_1/2 + j_2/4 + \cdots + j_n/2^n)} |1\rangle \right) |j_2 \cdots j_n\rangle = \frac{1}{2^{1/2}} \left( |0\rangle + e^{2\pi i 0.j_1 j_2 \cdots j_n} |1\rangle \right) |j_2 \cdots j_n\rangle, \quad (24)$$

where $|\xi_1\rangle$ is an intermediate quantum state. We then apply the Hadamard gate to the second qubit and repeat the controlled operation of $R_k$ for $k = 3, \ldots, n$ on the second qubit giving

$$|\xi_2\rangle = \frac{1}{2^{2/2}} \left( |0\rangle + e^{2\pi i 0.j_1 j_2 \cdots j_n} |1\rangle \right) \left( |0\rangle + e^{2\pi i 0.j_2 j_3 \cdots j_n} |1\rangle \right) |j_3 \cdots j_n\rangle. \quad (25)$$

We continue in this fashion for each qubit until $k = n$ and perform the SWAP operation to change the location of the least significant bit, which leads us to the final quantum state

$$|\xi_3\rangle = \frac{\left( |0\rangle + e^{2\pi i 0.j_n} |1\rangle \right) \left( |0\rangle + e^{2\pi i 0.j_{n-1} j_n} |1\rangle \right) \cdots \left( |0\rangle + e^{2\pi i 0.j_1 j_2 \cdots j_n} |1\rangle \right)}{2^{n/2}}. \quad (26)$$

Fig. 1 shows the circuit demonstration of the QFT. In the first step, one Hadamard gate and $n-1$ conditional rotation gates make a total of $n$ gates. Similarly, in the second step, we require $n - 1$ gates. Continuing in this manner, the total number of gates is $n + (n - 1) + \cdots + 1 = n(n + 1)/2$. Therefore, the computational complexity of the QFT is $O(n^2)$. In contrast, representing the same amount of information on a classical computer requires $N = 2^n$ classical bits. Therefore, implementing the discrete Fourier transform using the classical fast Fourier transform (FFT) would require a computational complexity of $O(n2^n)$. The QFT is often used as a subroutine of several quantum algorithms such as quantum phase estimation, which we discuss in the next section.

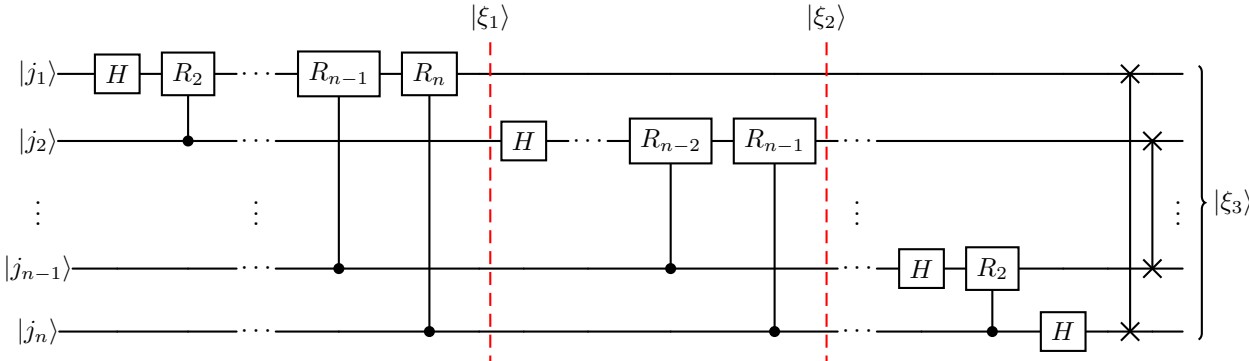

Figure 1: Circuit diagram to implement the quantum Fourier transform on a quantum computer.

### 4.2 Quantum phase estimation

Given a unitary operator $U$ acting on an $n$ qubit quantum state $|\phi\rangle$ with eigenvalue equation $U|\phi\rangle = e^{2\pi i \varphi}|\phi\rangle$, the objective of the quantum phase estimation (QPE) algorithm is to estimate the value of $\varphi$ (Kitaev, 1995). The QPE starts with a unitary transformation that can implement powers of $U$ conditioned on ancillary qubits

$$\frac{1}{2^{n/2}} \sum_{k=0}^{2^n-1} |k\rangle |\phi\rangle \rightarrow \frac{1}{2^{n/2}} \sum_{k=0}^{2^n-1} |k\rangle U^k |\phi\rangle. \tag{27}$$

The action of the unitary operator $U^k$ on the eigenvector $|\phi\rangle$ yields the eigenvalue $e^{2\pi i k \varphi}$ given as

$$\frac{1}{2^{n/2}} \sum_{k=0}^{2^n-1} |k\rangle U^k |\phi\rangle = \frac{1}{2^{n/2}} \sum_{k=0}^{2^n-1} |k\rangle \left( e^{2\pi i k \varphi} |\phi\rangle \right)$$

$$= \frac{1}{2^{n/2}} \sum_{k=0}^{2^n-1} e^{2\pi i k \varphi} |k\rangle |\phi\rangle.$$

Applying the inverse QFT to the equation above results in the eigenvalue estimate being stored in the quantum register as follows

$$\frac{1}{2^{n/2}} \sum_{k=0}^{2^n-1} e^{2\pi i k \varphi} |k\rangle |\phi\rangle \xrightarrow{QFT^\dagger} |\hat{\varphi}\rangle |\phi\rangle. \tag{28}$$

Quantum registers consist of multiple qubits that are used in quantum computers, similar to classical registers in classical computers (Nielsen & Chuang, 2011). We then perform a measurement in the computational basis to obtain the estimate $\hat{\varphi}$ of the eigenvalue.

The binary string $l$ with the highest success probability corresponds to the exact estimate $\hat{\varphi}$ only if $\hat{\varphi} = \frac{l}{2^n}$. Generally, $\varphi$ cannot be represented exactly by $\frac{l}{2^n}$, however, the QPE algorithm still gives the best possible approximation, according to Kitaev (1995), given by

$$\frac{1}{2^{n/2}} \sum_{k=0}^{2^n-1} e^{2\pi i k \varphi} |k\rangle |\phi\rangle \xrightarrow{QFT^\dagger} \frac{1}{2^{n/2}} \sum_{k=0}^{2^n-1} e^{2\pi i k (\varphi - j/2^n)} |i\rangle |\phi\rangle. \tag{29}$$

The highest success probability of measuring the closest estimate using the QPE algorithm becomes

$$p(j) = \left| \frac{1}{2^{n/2}} \sum_{k=0}^{2^n-1} e^{2\pi i k (\varphi - j/2^n)} \right|^2. \tag{30}$$

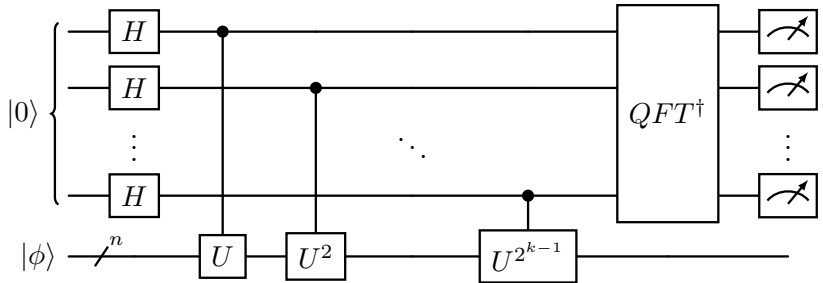

Figure 2: Circuit diagram to implement the quantum phase estimation on a quantum computer.

This depends on the difference between $\varphi$ and the binary fractional integer $j/2^n$. Therefore, to increase the precision of the estimate $\hat{\varphi}$, we need to add more ancilla qubits to the QFT. Fig. 2 presents the circuit of the quantum phase estimation algorithm.

QPE requires the implementation of powers of the unitary operator $U = e^{i\tilde{H}t}$ on a quantum computer using Hamiltonian simulation of the $\tilde{H}$ operator (Lloyd, 1996). Standard methods to implement Hamiltonian simulation involve decomposing the unitary matrix into all possible combinations of Pauli operators which has a huge computational complexity (Nielsen & Chuang, 2011). To obtain practical advantages over classical systems, we have to find solutions in which we can efficiently simulate the unitary system. Several algorithms have been proposed in this area, for example, Berry et al. (2007) consider a sparse Hamiltonian operator, another approach approximates it using Taylor series (Berry et al., 2015), and the commonly used Suzuki-Trotter method (Suzuki, 1976). Hamiltonian simulation can be efficiently done when the input state and the Hamiltonian operator are the same (Lloyd et al., 2014), with a method called density matrix exponentiation. It can be used in the quantum principal component analysis (qPCA) if the quantum density operator is non-sparse but low-rank. In this paper, we utilize this concept of qPCA for the Gaussian process quadrature rule.

### 4.3 Quantum principal component analysis

Instead of explicitly calculating the unitary matrix $U = e^{i\tilde{H}t}$, we can simulate the unitary operation using SWAP operators. For this, we add the extra condition that the Hermitian operator should be represented as the density matrix of some quantum state, that is, $\tilde{H} = \tilde{\rho}$. We apply the SWAP operation to the state $\tilde{\rho} \otimes \tilde{\sigma}$ and perform a partial trace operation. The expression for this simulation turns out to be (Lloyd et al., 2014; Schuld & Petruccione, 2021)

$$\text{tr}_2\{e^{-iS\Delta t}\left(\tilde{\sigma} \otimes \tilde{\rho}\right)e^{iS\Delta t}\} = \tilde{\sigma} - i\Delta t[\tilde{\rho}, \tilde{\sigma}] + O(\Delta t^2)$$
$$\approx e^{-i\tilde{\rho}\Delta t}\tilde{\sigma}e^{i\tilde{\rho}\Delta t},$$

where $\tilde{\sigma} - i\Delta t[\tilde{\rho}, \tilde{\sigma}]$ with $[\tilde{\rho}, \tilde{\sigma}] = \tilde{\rho}\tilde{\sigma} - \tilde{\sigma}\tilde{\rho}$ are the first terms of Taylor expansion of $e^{-i\tilde{\rho}\Delta t}\tilde{\sigma}e^{i\tilde{\rho}\Delta t}$, and $\text{tr}_2(.)$ is partial trace over second subsystem.

Density matrix exponentiation is often used with the quantum phase estimation technique. Once we have exponentiated the density matrix, we can extract the eigenvalues through quantum phase estimation. To do so, we need to prepare the powers of $U^k = e^{i\tilde{\rho}k\Delta t}$.

Lloyd et al. (2014) showed that using the exponential density matrix along with the estimation of the quantum phase, $O(\epsilon^{-3})$ copies of $\tilde{\rho}$ are required for this, where $\epsilon$ represents the precision to estimate the eigenvalues. These copies are then combined with an ancilla register of $n$ qubits in superposition to obtain

$$\rho_{\xi_4} = \sum_{k=0}^{2^n-1} |k\rangle\langle k| \otimes \tilde{\sigma} \otimes \tilde{\rho}^{(1)} \otimes \tilde{\rho}^{(2)} \otimes \cdots \otimes \tilde{\rho}^{(2^n)}. \tag{31}$$

We now perform a sequence of two-qubit conditional SWAP operators, each of which swaps the first qubit state $\tilde{\sigma}$ with the $g$th copy of $\tilde{\rho}$ conditioned on the ancilla qubit $|k\rangle$. The SWAP operators are entangled with the ancillary register, so the index $|k\rangle$ governs the sequence of SWAP operators up to the copy $\tilde{\rho}^{(k)}$ as

$$\rho_{\xi_5} = \frac{1}{K} \sum_{k=0}^{2^n-1} |k\Delta t\rangle \langle k\Delta t| \otimes \prod_{g=1}^{k} e^{-iS_g t}. \tag{32}$$

After taking the partial trace over all the copies of $\tilde{\rho}$, we obtain the result

$$\rho_{\xi_6} = \sum_{k=0}^{2^n-1} |k\rangle \langle k| \otimes e^{-ik\tilde{\rho}\Delta t} \tilde{\sigma} e^{ik\tilde{\rho}\Delta t} + O(\Delta t^2). \tag{33}$$

This mixed quantum state representation is suitable for applying the QFT (Schuld & Petruccione, 2021). Letting $\tilde{\sigma} = \tilde{\rho}$, it calculates the eigenvalues and eigenvectors of the operator $\tilde{\rho}$. Obtaining eigenvalues and eigenvectors in this way is known as quantum principal component analysis.

The SWAP operator is a sparse matrix and can be simulated efficiently. Lloyd et al. (2014) showed that this algorithm can be implemented with a computational complexity of $O(\log N)$ on a quantum computer when multiple copies of the density operator are already given as quantum states. This qPCA is qubit efficient as long as we do not have to estimate the eigenvalue to a precision that does not grow exponentially. According to Lloyd et al. (2014), this technique is only suitable when the density matrix is dominated by a few eigenvalues that do not require high precision for estimation.

## 5 Quantum Hilbert space low-rank Gaussian process quadrature

In this section, we provide the main contribution of our paper, which explores how quantum computers can be used to evaluate integrals using the Hilbert space low-rank Gaussian process quadrature. For this, we implement the Hilbert space kernel approximation developed by Farooq et al. (2024) for quantum-assisted Gaussian process regression and extend it to the GPQ rule. We start with the data matrix $\mathbf{X} \in \mathbb{R}^{N \times M}$ computed on a classical computer. This data matrix is exactly the same as used in Eqs. (11) and (12). We initially consider the amplitude encoding scheme in Appendix A.4 to encode the data matrix $\mathbf{X}$ with entries $x_n^m$ in the form

$$|\psi_{\mathbf{X}}\rangle = \sum_{m=0}^{M-1} \sum_{n=0}^{N-1} x_n^m |m\rangle |n\rangle. \tag{34}$$

The state $|\psi_{\mathbf{X}}\rangle$ has to be normalized, then, the entries $x_n^m$ satisfy the condition $\sum_{n,m} \|x_n^m\|^2 = 1$. The amplitude data encoding implicates a computational complexity $O(NM)$ in standard quantum state preparation methods (Schuld & Petruccione, 2021). However, efficient quantum amplitude encoding methods have been recently proposed, achieving a $O(\text{poly}(\log(NM)))$ computational complexity (Nakaji et al., 2022).

Starting from the quantum state in Eq. (34), we consider its Gram-Schmidt decomposition (Schuld et al., 2016) which gives

$$|\psi_{\mathbf{X}}\rangle = \sum_{r=1}^{R} \lambda_r |v_r\rangle |u_r\rangle. \tag{35}$$

The Gram-Schmidt decomposition works as a quantum analog of the SVD performed in (14). Further, consider the density matrix $\rho_{\mathbf{X}^\top \mathbf{X}} = \text{Tr}_n\{|\psi_{\mathbf{X}}\rangle \langle\psi_{\mathbf{X}}|\}$ given by tracing out the $|n\rangle$ register of the quantum state in Eq. (34). The resulting density matrix can be written in terms of the Gram-Schmidt decomposition in Eq. (35) as

$$\rho_{\mathbf{X}^\top \mathbf{X}} = \text{Tr}_n\{|\psi_{\mathbf{X}}\rangle \langle\psi_{\mathbf{X}}|\} = \sum_{r=1}^{R} \lambda_r^2 |v_r\rangle \langle v_r|. \tag{36}$$

Once the density operator is built, it is possible to apply it as an exponential evolution operator to the $|m\rangle$ register of $|\psi_{\mathbf{X}}\rangle$ following the qPCA technique (Lloyd et al., 2014) explained in Section 4.3. Since the state

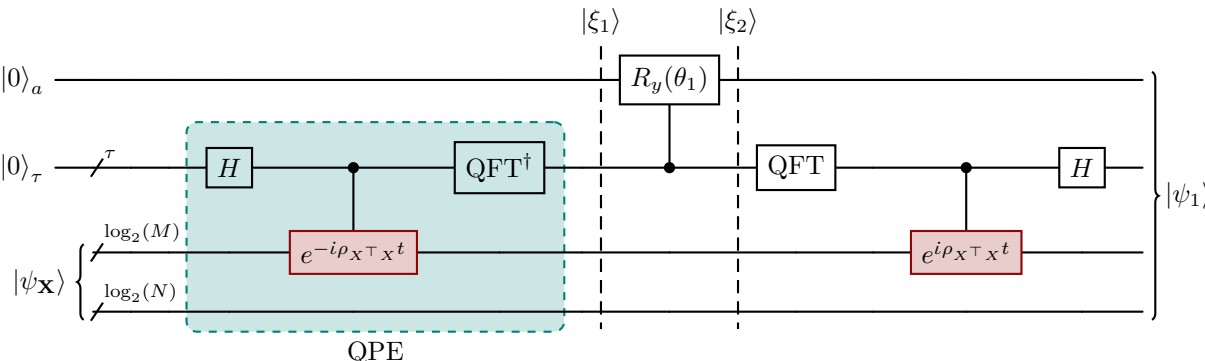

Figure 3: Quantum circuit to prepare the $|\psi_1\rangle$ quantum state. The red gates represent the parts of the algorithm where the qPCA is implemented, meanwhile, the green part envelopes the implementation of the QPE algorithm.

$|\psi_\mathbf{X}\rangle$ is the eigenstate of the operator in Eq. (36), it is also the eigenstate of $\exp(-i\rho_{\mathbf{X}^\top\mathbf{X}}t)$. With this argument, it is possible to implement the QPE algorithm (see Section 4.2) to estimate the eigenvalues of $\rho_{\mathbf{X}^\top\mathbf{X}}$.

The algorithm starts by building a specific state $|\psi_1\rangle$ that stores the information of the evaluation points. Fig. 3 shows the quantum circuit used to create this state. Four quantum registers are needed, the last two are the $|n\rangle_n$ and $|m\rangle_m$ registers needed to store $|\psi_\mathbf{X}\rangle$; then, we need a register $|0\rangle_\tau$ that will store the eigenstates; finally, we need an ancilla register $|0\rangle_a$. From Fig. 3, we can see that the ancilla register contains only one qubit, the second register contains $\tau$ qubits, and the last two registers together contain a total of $\log(NM)$ qubits.

Conditioning on $|0\rangle_\tau$, we implement the QPE algorithm with unitary evolution performed through qPCA. Denoting intermediate states as $|\xi_i\rangle$, the resulting state after the QPE algorithm is

$$|\xi_1\rangle = \sum_{r=1}^{R} \lambda_r |v_r\rangle_m |u_r\rangle_n |\lambda_r^2\rangle_\tau |0\rangle_a. \tag{37}$$

The binary representation of the eigenvalue $\lambda_r^2$ can be used to condition a $R_y(\theta_1)$ rotation around the $y$ axis of angle $\theta_1 = 2\arcsin\left(\frac{c_1}{\lambda_r^2+\sigma^2}\right)$ in the ancilla register. The eigenvalue estimation and the conditioned rotation correspond to the matrix inversion in (9). After the rotation, the state of the quantum circuit takes the form

$$|\xi_2\rangle = \sum_{r=1}^{R} \lambda_r |v_r\rangle_m |u_r\rangle_n |\lambda_r^2\rangle_\tau \left[\sqrt{1-\left(\frac{c_1}{\lambda_r^2+\sigma^2}\right)^2}|0\rangle_a + \left(\frac{c_1}{\lambda_r^2+\sigma^2}\right)|1\rangle_a\right]. \tag{38}$$

The term $\frac{c_1}{\lambda_r^2+\sigma^2}$ has to be smaller than one, this condition is satisfied by an appropriate selection of the constant $c_1$ (Cleve et al., 1998). Finally, it is possible to reverse the operation over the $\tau$ register, which provides the state

$$|\psi_1\rangle = \sum_{r=1}^{R} \lambda_r |v_r\rangle_m |u_r\rangle_n |0\rangle_\tau \left[\sqrt{1-\left(\frac{c_1}{\lambda_r^2+\sigma^2}\right)^2}|0\rangle_a + \frac{c_1}{\lambda_r^2+\sigma^2}|1\rangle_a\right]. \tag{39}$$

It is important to note that when the ancilla qubit is in the state $|1\rangle_a$, the state of the qubit registers $m$ and $n$ represents part of the quadrature expected value.

We can estimate the quadrature expected value $Q_{\text{BQ}}$ with quantum states by introducing the information of the vector $\mathbf{X}_\mu$ and the evaluation points $\mathbf{y}$. For this, we need an additional quantum state named $|\psi_2\rangle$

with the same number of registers as in $|\psi_1\rangle$. By considering the quantum states $|\mathbf{X}_\mu\rangle_m = \sum_m x_\mu^m |m\rangle$ and $|\mathbf{y}\rangle_n = \sum_n y_n |n\rangle$, we use the amplitude encoding scheme to build the state $|\psi_2\rangle = |\mathbf{X}_\mu\rangle_m |\mathbf{y}\rangle_n |0\rangle_\tau |1\rangle_a$, which is normalized. Note the ancilla qubit in the state $|1\rangle_a$ for $|\psi_2\rangle$. This ensures that when we perform the dot product between $\langle\psi_1 \mid \psi_2\rangle$, we always obtain the amplitude coefficient corresponding to the ancilla state $|1\rangle_a$ of $|\psi_1\rangle$ as the dot product $\langle 0|1\rangle_a$ is always zero.

Consider the dot product between the $|\psi_1\rangle$ and $|\psi_2\rangle$ states, which takes the form

$$\langle\psi_1 \mid \psi_2\rangle = c_1 \sum_{r=1}^{R} \frac{\lambda_r}{\lambda_r^2 + \sigma^2} \langle\mathbf{X}_\mu \mid v_r\rangle_m \langle\mathbf{y} \mid u_r\rangle_n. \tag{40}$$

Dividing the dot product by the rotation constant $c_1$, the result corresponds to the expected value $Q_{\text{BQ}} = \langle\psi_1 \mid \psi_2\rangle / c_1$ given in Eq. (14). The dot product between two states can be implemented in a quantum computer through a Hadamard test, which is explained in the Appendix A.5. As a result, the expected value in the quadrature can be written in terms of the probability $p_0$ of the ancilla qubit to be in the state $|0\rangle$ as

$$Q_{\text{BQ}} = \frac{2p_0 - 1}{c_1}. \tag{41}$$

The variance of the quadrature can be calculated using a similar circuit. After the quantum phase estimation, the circuit is in the intermediate state $|\xi_1\rangle$, now the conditioned rotation $R_y(\theta_2)$ has a different angle $\theta_2 = 2\arcsin\left(\frac{c_2}{\lambda_r\sqrt{\lambda_r^2 + \sigma^2}}\right)$. This results in the state

$$|\xi_3\rangle = \sum_{r=1}^{R} \lambda_r |v_r\rangle_m |u_r\rangle_n |\lambda_r^2\rangle_\tau \left[\sqrt{1 - \left(\frac{c_2}{\lambda_r\sqrt{\lambda_r^2 + \sigma^2}}\right)^2} |0\rangle_a + \left(\frac{c_2}{\lambda_r\sqrt{\lambda_r^2 + \sigma^2}}\right) |1\rangle_a\right], \tag{42}$$

where the constant $c_2$ satisfies the condition $c_2 \leq \lambda_r\sqrt{\lambda_r^2 + \sigma^2}$. The algorithm proceeds by conditioning the ancilla register to be in the state $|1\rangle_a$ and including the vector $\mathbf{X}_\mu$. In this case, we need an additional register, with the same length as $|m\rangle$, to store the quantum state $|\psi_2'\rangle = |\mathbf{X}_\mu\rangle$. By including an additional ancilla register $|0\rangle_b$ a SWAP test can be implemented between the $|\mathbf{X}_\mu\rangle$ and the $|m\rangle$ register of the circuit. Appendix A.6 provides the detailed steps for implementing the SWAP test on quantum computers. Measuring both ancillary qubits to be in the quantum state $|1\rangle$, the probability is

$$p_{11} = \frac{c_2^2}{2} \left(\sum_{r=1}^{R} \frac{1}{\lambda_r^2 + \sigma^2} - \sum_{r=1}^{R} \frac{1}{\lambda_r^2 + \sigma^2} |\langle\mathbf{X}_\mu \mid v_r\rangle|^2\right), \tag{43}$$

where the second term is proportional to the variance of the quadrature given in Eq. (15), then it can be written as

$$V_{\text{BQ}} = \sigma^2 \left(\sum_{r=1}^{R} \frac{1}{\lambda_r^2 + \sigma^2} - \frac{2p_{11}}{c_2^2}\right), \tag{44}$$

giving the Gaussian process quadrature variance. The quantum circuit to estimate the quadrature variance is shown in Fig. 4. Once the quantum computational method for Gaussian quadrature computation has been constructed we are interested in the computational complexity of the algorithm with respect to the classical Gaussian process quadratures. The quantum state preparation subroutines in our algorithm can be implemented using the quantum state preparation technique described by Nakaji et al. (2022), which efficiently prepares the quantum state $|\psi_\mathbf{X}\rangle$ with a computational complexity of $O(\text{poly}(\log(NM)))$ when the dataset consists of real numbers. We then apply qPCA along with QPE, which has a total computational cost $O(\log(M)\epsilon^{-3})$ (Schuld et al., 2016). The next step involves the conditional unitary, which has a low computational complexity of $O(\log(\frac{1}{\epsilon}))$, so we can neglect its computation cost. We perform measurements in the variance circuit before the SWAP test, which can be implemented with amplitude amplification techniques in $O(\kappa^2)$. We then employ the Hadamard and SWAP tests, whose complexity is linear in the

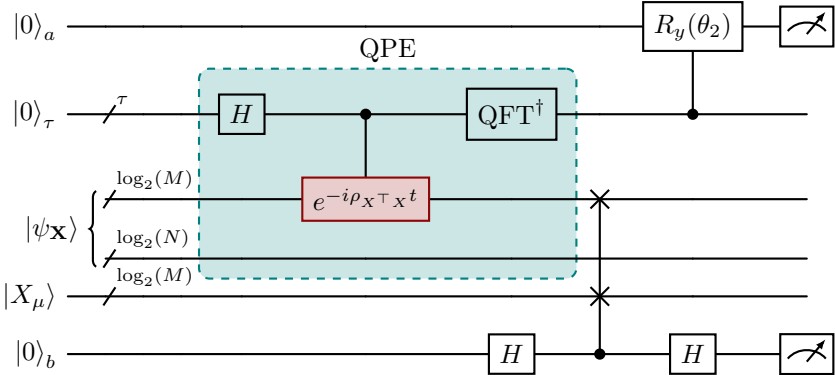

Figure 4: Quantum circuit to estimate the quadrature variance. The red gates represent the parts of the algorithm where the qPCA is implemented, meanwhile, the green part envelopes the implementation of the QPE algorithm.

number of qubits; thus, the measurement accounts for only a constant factor, which can be ignored. We use two separate quantum circuits, both of which have the same computational complexity and only introduce a constant multiplicative factor of 2 to the total computational complexity, which can be ignored, but allows us to implement more qubits in the numerical simulations. Therefore, the total computational cost to estimate the expected value and variance of our algorithm for the quantum Gaussian process quadrature rule is $O(\text{poly}(\log(NM))\log(M)\kappa^2\epsilon^{-3})$. On the other hand, the classical version of the GPQ rule requires a computational cost of $O(M^3)$, which shows that our algorithm is polynomially faster than the classical algorithm.

## 6 Numerical Simulations

The simulations performed in this paper correspond to executions of quantum circuits in a classical computer that simulate the result that would be obtained on quantum hardware. The classical simulations already pose several challenges for different factors. The QPE algorithm needs several qubits to provide reasonable estimates of the eigenvalues. Moreover, since we estimate multiple eigenvalues in this step, it is important to differentiate them properly. Implementing the qPCA evolution operator is sensitive to the parameter $t$. In this case, we select $t = 2\pi/\delta$ and modify the value of $\delta$ as suggested by Cleve et al. (1998), following the inequality $\delta > \lambda_{max}^2$ where $\lambda_{max}$ is the largest eigenvalue of the decomposition. For a good approximation, $\delta$ should be slightly greater than $\lambda_{max}^2$.

As an example for simple demonstration purposes, we considered the integral $\mathcal{I}_1 = \int_\Omega f(x)\mu(x)dx$ of a simple one-dimensional sinusoidal function $f(x) = 1 + \sin(x)$, similar to the one used for regression by Farooq et al. (2024), integrated over the interval $\Omega = [-\pi, \pi]$ with weight function $\mu(x) = 1$. We choose $\zeta$ within the same range $[-L, L]$ with $L = \pi$ to implement the framework outlined in this work. In this domain, the eigenfunctions corresponding to the Laplace operator are the sinusoidal functions $\phi_j(x) = L^{-1/2}\sin(\pi j(x + L)/2L)$, each associated with its respective eigenvalue $\lambda_j = (\pi j/2L)^2$.

The first demonstration of the algorithm was implemented using a constant value of $\delta = \lambda_{max}^2 + 0.01$, $N = 8$ evaluation points, $\tau = 16$ qubits to store the eigenvalues, $M = 4$ eigenfunctions to approximate the kernel and $10^6$ shots per circuit execution. The simulations were implemented using a classical simulator of a quantum computer using the QISKIT library (Qiskit contributors, 2023). For the simulations, we consider the square exponential kernel in Eq. (3) with hyperparameters $\sigma_f = 1$ and $\ell = 1$, whose spectral density is $S(\omega) = \sigma_f^2\sqrt{2\pi}l\exp\left(-\frac{l^2\omega^2}{2}\right)$. The noise $\varepsilon$ of the evaluation points has a Gaussian distribution with zero mean and variance $\sigma = 0.05$.

Fig. 5 shows the distribution that estimates the integral $\mathcal{I}_1$ given by the GPQ in Eqs. (6) and (7), the low-rank Hilbert space quadrature in Eqs. (9) and (10) and our proposed quantum Hilbert space quadrature

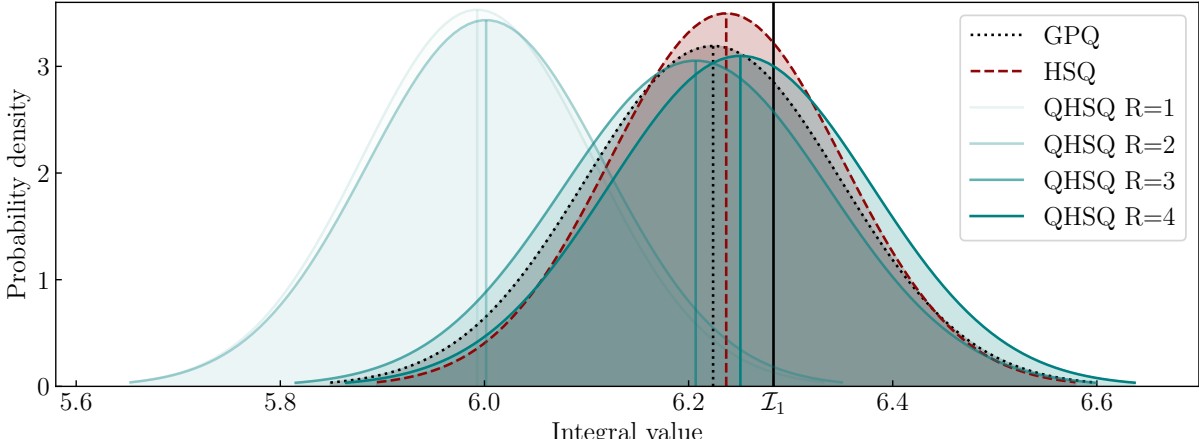

Figure 5: Estimate of the quadrature for the integral $\mathcal{I}_1$. The estimation results deliver the expected value and variance of the Gaussian distributions plotted above that approximates the integral. The dotted black plot corresponds to the GPQ, the dashed red plot corresponds to the HSQ and the solid green plots correspond to the estimates using our QHSQ algorithm with $R = 1, 2, 3, 4$.

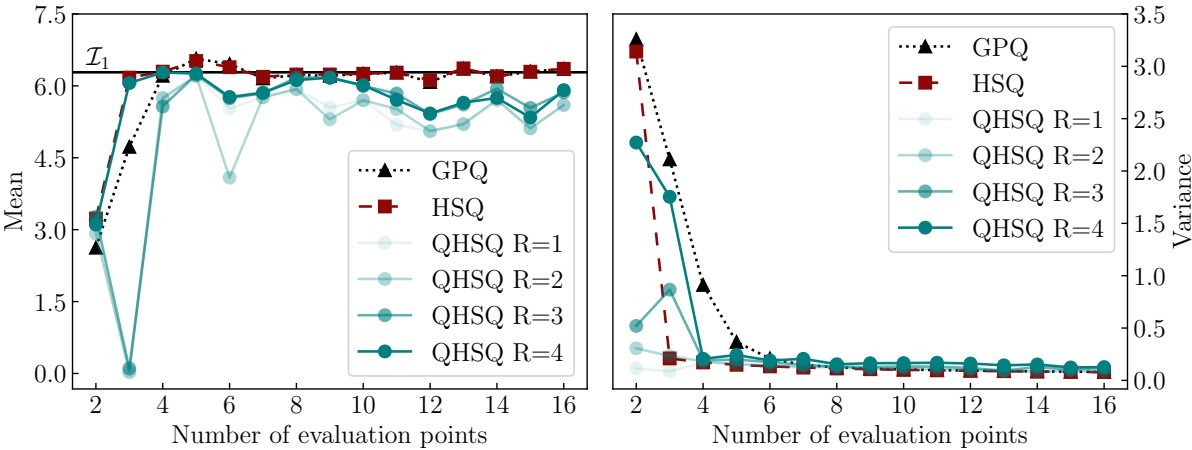

Figure 6: Expected value and variance of the Gaussian quadrature methods against the number of evaluation points. The dotted black line corresponds to the GPQ, the dashed red to the HSQ and the solid green lines correspond to our proposed QHSQ method with $R = 1, 2, 3, 4$.

method. The distributions given by $R = 1, 2$ are quite distant from the integral value, meaning those low-rank approximations do not have enough information to approximate the integral. Also, these two distributions have a similar expected value, which is expected since the vector $X_\mu$ of this problem has $X_{\mu_2} = X_{\mu_4} = 0$. Similarly, the distributions with $R = 3, 4$ have a similar expected value, this time around the expected value of the integral given by the classical HSQ, showing the effectiveness of our proposed algorithm.

The expected value and variance of the method were plotted in Fig. 6 against the number of evaluation points used for the quadrature, which are symmetrically distributed around $x = 0$. Besides the number of evaluation points, all simulation parameters are the same as in the previous example. The implementation of the algorithm is susceptible to the parameter $\delta$, occasionally, it can lead to wrong results due to the small rotations implemented in the circuit and the limited number of $\tau$ qubits used to approximate the eigenvalues

$\lambda_r^2$. To avoid outlier results, we implemented the algorithm with six different values of $\delta = \lambda_{max}^2 + \epsilon$, with $\epsilon \in \{0.01, 0.009, 0.008, 0.007, 0.006\}$, and we took the median of these results for each number of evaluation points.

For the quantum implementation, it can be seen that as $R$ increases, the expected value approaches the integral real value and also the classical counterpart of the method. It can also be seen that the quantum method has greater fluctuation than the classical methods, mainly attributed to the limited amount of qubits used in the simulations, the finite amount of shots, and the small rotations implemented in the algorithm. On the other hand, we can see how the variance of the quadratures decreases with the number of evaluation points as expected.

## 7    Conclusion

In this paper, we introduced a novel quantum algorithm for low-rank Gaussian process quadrature. The method combines the uncertainty quantification properties of Bayesian quadrature methods for numerical integration (O'Hagan, 1991; Minka, 2000) with the speedups provided by quantum computation. Our method provides a polynomial complexity reduction compared to the classical GPQ rule, enhancing the solution of analytically intractable integration problems. We have also demonstrated the performance of the method using numerical simulations run on a classical simulator of a quantum computer, which shows that the algorithm works in practice. Our quantum algorithm computes the same solution as obtained from a classical computer but with accelerated speed. The simulation results slightly deviate from the classical GPQ because we use an approximate implementation of the unitary operation for qPCA along with a limited number of qubits.

The proposed quantum algorithm is suitable for a fault-tolerant quantum computer with enough quantum registers to implement the algorithm precisely. Recent developments in materials research have shown promise in the realization of fully scalable quantum computers with higher coherence times (Acharya et al., 2024), which opens a window for implementing the method in real hardware. Also, we discussed the sensitivity of the method concerning the QPE $\delta$ parameter. However, this drawback can be overcome with advanced quantum simulation techniques (Zulehner & Wille, 2019) that improve the classical demonstration of the method.

Future work could include the optimization of the quantum circuit needed for the algorithm implementation and demonstrations using advanced classical simulation techniques for quantum computers. Besides the well-known data encoding problem in quantum computing, the complexity of the algorithm is mainly dominated by the QPE and the qPCA parts of the algorithm. These steps can be reduced using iterative approaches for the QPE algorithm (Smith et al., 2022), allowing it to be parallelized. The qPCA implementation can be enhanced using a hybrid approach, despite increasing the classical resources needed to implement the algorithm, a variational circuit can be implemented to reduce the complexity of the quantum circuit that is executed (Xin et al., 2021). This could enable the algorithm to be executed in a noisy intermediate-scale quantum computer.

### Author Contributions

C.A. Galvis-Florez and A. Farooq contributed equally to this work. All the authors contributed to the original idea, writing, and finalizing the paper results.

### Acknowledgments

C.A. Galvis-Florez, A. Farooq and Simo Särkkä thank the Research Council of Finland for funding this research (project 350221).

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

## A Review of Fundamental Concepts of Quantum Computing

In this section, we will review the fundamental concepts used to implement our algorithm, known as the quantum-assisted Hilbert space Gaussian process quadrature rule. We follow and use the symbols and notations as used by Schuld & Petruccione (2021).

### A.1 Qubits

The basic unit of a quantum computer is the qubit, which is the analog of the bit in classical computers. A single qubit can be either in quantum state $|0\rangle = \begin{bmatrix} 1 \\ 0 \end{bmatrix}$ or $|1\rangle = \begin{bmatrix} 0 \\ 1 \end{bmatrix}$. Moreover, it can also be represented as a linear combination of both states, known as superposition, given by (Nielsen & Chuang, 2011)

$$|q\rangle = \alpha |0\rangle + \beta |1\rangle, \tag{45}$$

with $\alpha, \beta \in \mathbb{C}$. Moreover, the squared amplitudes $|\alpha|^2$ and $|\beta|^2$ correspond to the probability of measuring the qubit $|q\rangle$ in the state $|0\rangle$ or $|1\rangle$ respectively. Then, they satisfy the condition $|\alpha|^2 + |\beta|^2 = 1$. An $n$-qubit unentangled quantum system can be represented as the tensor product of single-qubit states

$$|\psi\rangle = |q_1\rangle \otimes \cdots \otimes |q_n\rangle. \tag{46}$$

More generally, it can be written as the superposition

$$|\psi\rangle = \sum_{i=0}^{2^n-1} \alpha_i |i\rangle, \tag{47}$$

where $\alpha_i \in \mathbb{C}$, $\sum_{i=0}^{2^n-1} |\alpha_i|^2 = 1$, and $\{|i\rangle\}$ represents the computational basis $\{|0\cdots0\rangle = |0\rangle, \cdots, |1\cdots1\rangle = |2^n-1\rangle\}$. Interactions between qubits can lead them into entangled states, which play an important role in the development of quantum algorithms and applications of quantum technologies. Entangled states cannot be represented as the product of individual states, for example, an entangled two-qubit quantum state can be represented as

$$|\psi\rangle = \frac{1}{\sqrt{2}} \left( |0\rangle_{q_1} |0\rangle_{q_2} + |1\rangle_{q_1} |1\rangle_{q_2} \right) = \frac{1}{\sqrt{2}} \left( |00\rangle_{12} + |11\rangle_{12} \right). \tag{48}$$

The complete state cannot be written as $|q_1\rangle \otimes |q_2\rangle$, however, it corresponds to the superposition of specific states of the computational basis.

It is important to notice that neither qubit 1 nor 2 in the quantum state has a definite value, but when the state of one qubit is measured, a probabilistic process, then the state of the other qubit is completely defined. When the qubit $q_1$ is measured in a specific state, for example, $|0\rangle$, the qubit $q_2$ immediately takes the value $|0\rangle$. The same logic applies when the measurement is performed over the qubit $q_2$.

### A.2 Quantum gates

Quantum logic gates are the fundamental building blocks of quantum circuits, whose action on the quantum states defines the dynamics of the circuit. This section follows Schuld & Petruccione (2021) to give a brief introduction to single and multiqubit gates. Quantum gates are realized through unitary transformation.

Unlike the classical gates, quantum gates are reversible. For a single qubit, there are three fundamental gates known as Pauli matrices

$$\sigma_x = \begin{pmatrix} 0 & 1 \\ 1 & 0 \end{pmatrix}, \qquad \sigma_y = \begin{pmatrix} 0 & -i \\ i & 0 \end{pmatrix}, \qquad \sigma_z = \begin{pmatrix} 1 & 0 \\ 0 & -1 \end{pmatrix}. \tag{49}$$

The Pauli $X$-gate is equivalent to the classical NOT gate for a quantum system. This gate flips $|0\rangle$ to $|1\rangle$ and vice versa. This can be represented in matrix form as

$$\sigma_x |0\rangle = \begin{pmatrix} 0 & 1 \\ 1 & 0 \end{pmatrix} \begin{pmatrix} 1 \\ 0 \end{pmatrix} = \begin{pmatrix} 0 \\ 1 \end{pmatrix} = |1\rangle, \qquad \sigma_x |1\rangle = \begin{pmatrix} 0 & 1 \\ 1 & 0 \end{pmatrix} \begin{pmatrix} 0 \\ 1 \end{pmatrix} = \begin{pmatrix} 1 \\ 0 \end{pmatrix} = |0\rangle. \tag{50}$$

The Pauli $Z$-gate introduces a phase of -1 to the $|1\rangle$ state and keeps $|0\rangle$ state unchanged

$$\sigma_z |0\rangle = \begin{pmatrix} 1 & 0 \\ 0 & -1 \end{pmatrix} \begin{pmatrix} 1 \\ 0 \end{pmatrix} = \begin{pmatrix} 1 \\ 0 \end{pmatrix} = |0\rangle, \qquad \sigma_z |1\rangle = \begin{pmatrix} 1 & 0 \\ 0 & -1 \end{pmatrix} \begin{pmatrix} 0 \\ 1 \end{pmatrix} = \begin{pmatrix} 0 \\ -1 \end{pmatrix} = -|1\rangle. \tag{51}$$

Similarly, the Pauli $Y$ gate flips the quantum state and performs a phase flip operation on a quantum state

$$\sigma_y |0\rangle = \begin{pmatrix} 0 & -i \\ i & 0 \end{pmatrix} \begin{pmatrix} 1 \\ 0 \end{pmatrix} = \begin{pmatrix} 0 \\ i \end{pmatrix} = i|1\rangle, \qquad \sigma_y |1\rangle = \begin{pmatrix} 0 & -i \\ i & 0 \end{pmatrix} \begin{pmatrix} 0 \\ 1 \end{pmatrix} = \begin{pmatrix} -i \\ 0 \end{pmatrix} = -i|0\rangle. \tag{52}$$

A more general form of single-qubit gates is the rotations around the $x, y, z$ axis of the Bloch sphere Nielsen & Chuang (2011). These rotations are parametrized with respect to an angle and have the form

$$\begin{aligned} R_x(\theta) &= e^{-i\frac{\theta}{2}\sigma_x} = \begin{pmatrix} \cos(\frac{\theta}{2}) & -i\sin(\frac{\theta}{2}) \\ -i\sin(\frac{\theta}{2}) & \cos(\frac{\theta}{2}) \end{pmatrix}, \\ R_y(\theta) &= e^{-i\frac{\theta}{2}\sigma_y} = \begin{pmatrix} \cos(\frac{\theta}{2}) & -\sin(\frac{\theta}{2}) \\ \sin(\frac{\theta}{2}) & \cos(\frac{\theta}{2}) \end{pmatrix}, \\ R_z(\theta) &= e^{-i\frac{\theta}{2}\sigma_z} = \begin{pmatrix} e^{-i\frac{\theta}{2}} & 0 \\ 0 & e^{i\frac{\theta}{2}} \end{pmatrix}. \end{aligned} \tag{53}$$

The previous rotations can now defined a general rotation of a qubit parametrized by three angles given by

$$U(\theta, \phi, \lambda) = R_z(\phi)R_x(-\frac{\pi}{2})R_z(\theta)R_x(\frac{\pi}{2})R_z(\lambda) \doteq \begin{pmatrix} \cos\frac{\theta}{2} & -e^{-i\lambda}\sin\frac{\theta}{2} \\ e^{i\phi}\sin\frac{\theta}{2} & e^{i(\phi+\lambda)}\cos\frac{\theta}{2} \end{pmatrix}. \tag{54}$$

Besides single qubit gates, multiqubit gates act over multiple qubits through unitary quantum operations. These gates are usually responsible for entanglement between qubits. The Controlled-NOT (CNOT) gate is the most common two-qubit quantum gate. The quantum state is changed only when the control qubit is in the $|1\rangle$ state and remains unchanged if the control qubit is in the $|0\rangle$ state. The matrix representation of the Controlled-NOT gate is defined as follows

$$CNOT = |0\rangle\langle 0| \otimes I + |1\rangle\langle 1| \otimes X = \begin{pmatrix} 1 & 0 & 0 & 0 \\ 0 & 1 & 0 & 0 \\ 0 & 0 & 0 & 1 \\ 0 & 0 & 1 & 0 \end{pmatrix}. \tag{55}$$

The operation of the CNOT gate on two-qubit quantum states is given by

$$CNOT |00\rangle = |00\rangle, \ \ CNOT |01\rangle = |01\rangle, \ \ CNOT |10\rangle = |11\rangle, \ \ CNOT |11\rangle = |10\rangle. \tag{56}$$

The concept of controlled gates is extended to controlled general rotations, which can be represented in a matrix of the form

$$CU(\theta, \phi, \lambda) = |0\rangle\langle 0| \otimes I + |1\rangle\langle 1| \otimes U(\theta, \phi, \lambda) = \begin{pmatrix} 1 & 0 & 0 & 0 \\ 0 & 1 & 0 & 0 \\ 0 & 0 & \cos\frac{\theta}{2} & -e^{-i\lambda}\sin\frac{\theta}{2} \\ 0 & 0 & e^{i\phi}\sin\frac{\theta}{2} & e^{i(\phi+\lambda)}\cos\frac{\theta}{2} \end{pmatrix}. \tag{57}$$

A SWAP gate is often used in quantum algorithms to exchange qubits in two quantum registers. The SWAP gate is given by

$$SWAP = \begin{pmatrix} 1 & 0 & 0 & 0 \\ 0 & 0 & 1 & 0 \\ 0 & 1 & 0 & 0 \\ 0 & 0 & 0 & 1 \end{pmatrix}. \tag{58}$$

Acting over two arbitrary states $|\phi\rangle |\psi\rangle$ it will give

$$SWAP |\phi\rangle |\psi\rangle = |\psi\rangle |\phi\rangle. \tag{59}$$

We have reviewed some of the most common quantum gates, however, with one two-qubit gate and four single-qubit gates, is possible to build any multiqubit quantum circuit (DiVincenzo, 1995).

### A.3 Quantum measurements

We need to perform measurements to retrieve quantum information from quantum state $|\psi\rangle$. The outcomes of these measurements are random, whose probabilities are given by the elementary operators $\mathbf{M}_i = |\phi_i\rangle \langle\phi_i|$ through Born's rule (Nielsen & Chuang, 2011)

$$p_i = \langle\psi|\mathbf{M}_i|\psi\rangle. \tag{60}$$

The measurement operators are a collection of rank-1 projectors that sum to the identity $\sum_i \mathbf{M}_i = \mathbb{I}$. Generally, we take the computational basis, which is orthonormal, to define the measurement operators for different quantum algorithms.

Similarly, we can perform partial measurements over some fraction of the qubits that compose the system. For example, performing measurements only on the first qubit in the $|0\rangle$ basis gives

$$p_0 = \langle\psi| |0\rangle \langle0| \otimes \mathbb{I} \cdots \otimes \mathbb{I} |\psi\rangle. \tag{61}$$

Partial measurements are used in the Hadamard and SWAP tests to estimate the inner product between quantum states.

### A.4 Amplitude encoding

The amplitude encoding scheme encodes each element of the normalized vector into a coefficient of the quantum state. Suppose we have a vector $\mathbf{x} = [x_1, \cdots, x_{2^n}]$, where $\mathbf{x} \in \mathbb{C}^{2^n}$. We first normalize the vector such that $\sum_i^{2^n} |x_i|^2 = 1$ and encode the vector into the quantum state as (Schuld & Petruccione, 2021)

$$|\psi_{\mathbf{x}}\rangle = \sum_{i=0}^{2^n - 1} x_i |i\rangle. \tag{62}$$

We can also encode the matrix $\mathbf{A} \in \mathbb{C}^{2^n \times 2^n}$ with entries $a_{ij}$ that fulfill $\sum_{ij} \|a_{ij}\|^2 = 1$ in a similar fashion to

$$|\psi_{\mathbf{A}}\rangle = \sum_{i=0}^{2^n - 1} \sum_{j=0}^{2^n - 1} a_{ij} |i\rangle |j\rangle, \tag{63}$$

with the enlarged Hilbert space accordingly. The index registers $|i\rangle$, $|j\rangle$ refer to the $i$th row and the $j$th column of the matrix $\mathbf{A}$.

### A.5 Hadamard test

The Hadamard test calculates the inner product between two states along with correct signs. The additional overhead with the Hadamard test is that the ancilla qubit is entangled with both quantum states $|\psi_1\rangle$ and $|\psi_2\rangle$ as follows (Schuld & Petruccione, 2021):

$$|\psi_3\rangle = \frac{1}{\sqrt{2}} \left( |0\rangle |\psi_1\rangle + |1\rangle |\psi_2\rangle \right). \tag{64}$$

Once we have the state available in this entangled form, we then apply a Hadamard gate to the ancilla qubit, resulting in

$$|\psi_3\rangle = \frac{1}{2}|0\rangle \otimes (|\psi_1\rangle + |\psi_2\rangle) + \frac{1}{2} \otimes (|\psi_1\rangle - |\psi_2\rangle). \tag{65}$$

After this, we measure on the computational basis. The real and imaginary parts of the inner product between the two quantum states are given by

$$Re(\langle\psi_1|\psi_2\rangle) = 2p_0 - 1, \quad Im(\langle\psi_1|\psi_2\rangle) = 1 - 2p_0. \tag{66}$$

When the states are real, the calculation of the real part becomes the actual inner product between the two quantum states.

## A.6 SWAP test

The SWAP test in quantum computers is used to extract the square of the absolute value of the inner product between two quantum states $|\psi_1\rangle$ and $|\psi_2\rangle$ (Schuld & Petruccione, 2021). Suppose that we are given these two quantum states in the tensor product form $|\psi_1\rangle|\psi_2\rangle$. We start by adding an extra ancilla qubit $|0\rangle|\psi_1\rangle|\psi_2\rangle$. Then, we apply a Hadamard gate to the ancilla qubit, resulting in the following quantum state

$$\frac{1}{\sqrt{2}}(|0\rangle + |1\rangle)|\psi_1\rangle|\psi_2\rangle = \frac{1}{\sqrt{2}}(|0\rangle|\psi_1\rangle|\psi_2\rangle + |1\rangle|\psi_1\rangle|\psi_2\rangle). \tag{67}$$

Next, we apply the swap operation between the two states conditioned on the ancilla qubit being in the quantum state $|1\rangle$ as

$$\frac{1}{\sqrt{2}}(|0\rangle|\psi_1\rangle|\psi_2\rangle + |1\rangle|\psi_1\rangle|\psi_2\rangle) \xrightarrow[\text{SWAP}]{\text{Controlled}} \frac{1}{\sqrt{2}}(|0\rangle|\psi_1\rangle|\psi_2\rangle + |1\rangle|\psi_2\rangle|\psi_1\rangle). \tag{68}$$

Applying another Hadamard gate on the ancilla qubit we have

$$\frac{1}{2}(|0\rangle \otimes (|\psi_1\rangle|\psi_2\rangle + |\psi_2\rangle|\psi_1\rangle) + |1\rangle \otimes (|\psi_1\rangle|\psi_2\rangle - |\psi_2\rangle|\psi_1\rangle)). \tag{69}$$

When we measure the ancilla qubit, the probability of measuring the state $|0\rangle$ is

$$p_0 = \frac{1}{2} + \frac{1}{2}|\langle\psi_1|\psi_2\rangle|^2. \tag{70}$$

This method evaluates the inner product, however, the squared of the absolute value does not allow us to estimate the sign of the inner product.

