# OpenReview forum: "Provable Quantum Algorithm Advantage for Gaussian Process Quadrature"
_TMLR — Accepted by TMLR_

### Review · Reviewer_b6QV · 2024-10-11

**Summary Of Contributions:**

The paper develops a novel algorithm to perform numerical integration using Gaussian process quadrature rules on a quantum computer. After providing background on Hilbert space approximations for the Gaussian Process as well as the background on quantum algorithms, the paper develops the quantum Hilbert space low-rank Gaussian process quadrature and shows its reduction in complexity compared to methods on a classical computer.

**Audience:**

Yes

**Broader Impact Concerns:**

No Concerns

**Claims And Evidence:**

Yes

**Requested Changes:**

My comments mainly concern the readability of the paper.

- Maybe this is a naive question, but what is the intuitive reason for the reduced complexity of the quantum algorithm? I guess it has to do with the ability of quantum states to be in a superposition, but it might be helpful to spell this out for laymen like me.

- The introduction talks about complexity reduction without explaining kappa, s, and epsilon. In general, given all the notation, a small table at the beginning summarizing the most important symbols might be helpful. This table could focus mainly on symbols related to the quantum algorithm, since I think the audience of TMLR is more familiar with the notation used in Section 2.

- What exactly is different from the previous paper by Farooq et al. (2024)? It might be useful to flesh this out more.

- The |j> notation is extremely confusing to me, and while the appendix is helpful, I think this should be explained when it first appears in Section 3.1.

- I think equation (27) should be equation (26).

- Is it possible to make more references to section 2 in section 4 to try to make it as clear as possible which steps in the quantum algorithm roughly correspond to a calculation in calculating (14) and (15)?

**Strengths And Weaknesses:**

Strengths
-----------------

- The paper seems well written and illustrated
-  The idea and speedup for this important problem is interesting

Weaknesses
-----------------
- Despite the extensive background the paper is hard to understand, at least for me. I suspect
it will be similar for other researchers with ML/Stats background.
- I am not entirely sure how interesting this topic is for the TMLR audience, although there is one earlier paper in TMLR on the quantum advantage of Reinforcement Learning, so there appears to be some interest.
- I am not sure how much contribution the paper delivers compared to Farooq et al (2024), see the requested changes comment below.

---

> ### Author Response · Authors · 2024-11-29
> **Response reviewer b6QV 1/2**
>
> We would like to thank you for your insightful suggestions and comments, that helped us in improving the quality of our paper. Also, We thank the reviewer for pointing out the contribution of our paper. We have revised our manuscript by incorporating your suggestions and comments. Below is our response to your comments.
>
> **Response to Weaknesses:**
> *  This topic has indeed appeared in TMLR before. The contribution is clarified in response to requested change 3.
>
> **Response to the Requested changes:**
>
> **R1:** The complexity reduction advantage in quantum computing arises from the ability to represent information using the superposition of quantum states and to perform unitary operations to obtain the desired results. The quantum Fourier transform, quantum phase estimation, and quantum principal component analysis use the quantum superposition of states and perform unitary operations on them to provide advantages over their classical counterparts. Our algorithm is based on these quantum techniques, which reduces computational complexity compared to classical methods. As suggested by the reviewer, we have added the following text to the introduction:
> * Page 2, Second paragraph, Second line:
> Quantum computers exploit superposition, entanglement, and interference to perform a task, which substantially reduces the computational complexity of the algorithm (Nielsen & Chuang, 2011).
>
> * Page 3, Second paragraph:
> Our algorithm harnesses the quantum Fourier transform, quantum phase estimation, and quantum principal component analysis to exploit the superposition of quantum states for solving matrix inversion in GPQ, thereby providing an advantage over the classical approach.
>
> **R2:** We have revised our text to include definitions for all the missing variables. We have also included a table to define all the symbols used in our proposed algorithm.
> * Page 3, Third paragraph, Third line:
> $\kappa$ is the condition number, $s$ is the sparsity of the matrix, $\epsilon$ is the desired level of accuracy, and$\dots$
>
> * See table in Section 1.
> * Page 19, After equation (57):
> A SWAP gate is often used in quantum algorithms to exchange qubits in two quantum registers. The $\mathrm{SWAP}$ gate is given by
>
> $$
> \begin{align}
>     SWAP = \begin{pmatrix}
>             1&0&0&0 \\\\
>             0&0&1&0 \\\\
>             0&1&0&0 \\\\
>             0&0&0&1\\\\
>         \end{pmatrix}.
> \end{align}
> $$
>
> Acting over two arbitrary states $\mid\phi\rangle \mid\psi\rangle$ it will give
> $$SWAP \mid\phi\rangle \mid\psi\rangle = \mid\psi\rangle \mid\phi\rangle.$$
> 	We have reviewed some of the most common quantum gates, however, with one two-qubit gate and four single-qubit gates $\dots$
>
> **R3:** The paper by Farooq et al. (2024) considers the regression problem. In this paper, we develop a quantum Gaussian process quadrature method for probabilistic numerical integration. We have now included a bit more explanation on this in the paper.
> * Page 3, second paragraph:
> In this paper, we develop a low-rank quantum Gaussian process quadrature method for numerical integration using a low-rank Gaussian process prior, using the GPR method in Farooq. et. al (2024) as the starting point.
>
> **R4:** We revised the paper to explain the notation $\mid{j}\rangle$ when it first appears in Section 3.1, which is now Section 4.1. Additionally, we have included a reference to the appendix before starting the fundamentals of the quantum algorithm in the main text for readers better understanding who are unfamiliar with quantum notation.
> * Page 6, Before equation (17):
> The QFT acts over the basis element $\mid{j}\rangle$ as
> * Page 6, Section 4, Third line:
> We have provided more details on the representation of quantum states, unitary transformation operations, and quantum measurements in Appendices A.1, A.2, and A.3.
> * Page 6, After equation (19):
> To understand how the QFT has been implemented on quantum computers, it is helpful to express the state $\mid j \rangle$ in a binary representation as $j=j_{1}j_{2}\cdots j_{n}$, where $j_{1},j_{2},\cdots,j_{n}$ are either $0$ or $1$. For example, we can represent the state $\mid{j}\rangle$ when $j=2$ as a unit vector using binary representation: $\mid{2}\rangle=\mid{10}\rangle=\mid{1}\rangle\otimes \mid{0}\rangle=\begin{pmatrix}
>    0 & 0&1&0
> \end{pmatrix}^\top$, where single qubit states are $\mid{0}\rangle=\begin{pmatrix}
>    1\\\\0
> \end{pmatrix}$ and $\mid{1}\rangle=\begin{pmatrix}
>    0\\\\1
> \end{pmatrix}$.
>
> **R5:** The text preceding Equation (27), which is now Equation (28), may have caused some confusion, leading to the impression that Equation (28) should be Equation (27). We have carefully revised the text to clarify this point and eliminate any ambiguity.
> * Page 8, Before equation (28):
> Applying the inverse QFT to the equation above results in the eigenvalue estimate being stored in the quantum register as follows

---

> ### Author Response · Authors · 2024-11-29
> **Response reviewer b6QV 2/2**
>
> **R6:** We have added more references to section 2 in section 4 to make it more clear correspondence to a calculation in calculating  (14) and (15) using quantum algorithm.
> * Page 10, Section 5, First paragraph:
> We start with the data matrix $\mathbf{X} \in \mathbb{R}^{N \times M}$ computed on a classical computer. This data matrix is exactly the same as used in Equations (11) and (12).
> * Page 10, After equation (35):
>  The Gram-Schmidt decomposition works as a quantum analog of the SVD performed in (14).
> * Page 11, Before equation (38):
> The eigenvalue estimation and the conditioned rotation correspond to the matrix inversion in (9)

---

### Review · Reviewer_GMVh · 2024-11-06

**Summary Of Contributions:**

This paper provides the details on implementation of the Gaussian Process Quadrature (GPQ) on a quantum computer. The paper is well written and is clear about the achievements. The main contributions of this work are as follows:

1 - detailed implementation of the GPQ on a quantum computer using the phase estimation approach called QHSQ

2 - verification of the analytic results on small scale simulation

3 - detailed derivation of the whole GPQ

**Audience:**

Yes

**Claims And Evidence:**

Yes

**Requested Changes:**

I feel that while the paper is complete and self-contained, the transition between the existing HSQ to QHSQ was not straight forward. I would highly appreciate if there was a clear step by step outline of the classical vs the quantum algorithm in the paper.

Some more explanations of the steps of the algorithms would be very beneficial for understanding

**Strengths And Weaknesses:**

The implementation of the HSQ in quantum computer.
Integration of the input setting algorithm from Nakaji et al. is a nice complexity reduction.

Sometimes the used language is a bit unclear making the paper reading a bit difficult. For instance, page 9, "Conditioning on the second register, we implement...." should probably be for better understanding "Conditioning on the $\vert0\rangle_\tau$, we implement...."

According to the authors, when one of the resulting qubit register is $\vert1\rangle_a$ the results can be retrieved and contains the expected value. However what happens when the computation is finished and the indicator register is $\vert0\rangle_a$?

Does the computation of the expected value and of the variance require two separate experiments or can it be implemented as a part of the same quantum circuits?

---

> ### Author Response · Authors · 2024-11-29
> **Response reviewer GMVh**
>
> We would like to thank you for your insightful suggestions and comments. We are thankful to the reviewer for pointing out the contribution of our paper. We have revised our manuscript by incorporating your suggestions and comments. Below is our response to your comments.
>
> **Response to Weaknesses:**
>
> **R1:** We have corrected the language in the sentence as suggested.
>   * Page 9, after equation (31), second line:
> We now perform a sequence of two-qubit conditional SWAP operators, each of which swaps the first qubit state $\tilde{\sigma}$ with the $g$th copy of $\tilde{\rho}$ conditioned on the ancilla qubit $|{k}\rangle$
>   * Page 10, Before equation (37):
> Conditioning on $|{0}\rangle_\tau$, we implement
>
> **R2:** We missed providing details on this point. We have now included the necessary explanation in our revised version. When the ancilla register is $|{0}\rangle_a$, the dot product that is taking place between the states $|{\psi_1}\rangle$ and $|{\psi_2}\rangle$ will be zero. In other words, the probability of that register being in that state after the dot product is null. The algorithm is built that way since the information we need to get the expected value is stored in $|{1}\rangle_a$ (see Eq. 39).
> * Page 11, Paragraph before equation (40), 5 line:
> Note the ancilla qubit in the state $|{1}\rangle_a$ for $|{\psi_{2}}\rangle$. This ensures that when we perform the dot product between $\langle{\psi_1\mid\psi_2}\rangle$, we always obtain the amplitude coefficient corresponding to the ancilla state $|{1}\rangle_a$ of $|{\psi_{1}}\rangle$ as
> the dot product $\langle{0|1}\rangle_a$ is always zero.
>
> **R3:** We are thankful to the reviewer for this comment and for pointing out this. Both quantities are calculated using independent circuits, which could be executed in parallel on a bigger circuit by increasing the number of qubits. However, we consider independent circuits mainly because simulating them on a desktop computer is expensive, so independent circuits allowed us to consider more data points in the simulation. We added a line in the text regarding this.
> * Page 12, last paragraph, 11 line:
> We use two separate quantum circuits, both of which have the same computational complexity and only introduce a constant multiplicative factor of $2$ to the total computational complexity, which can be ignored, but allows us to implement more qubits in the numerical simulations.  Therefore, the total computational cost to estimate the expected value and variance of our algorithm for the quantum Gaussian process quadrature rule is $O(\operatorname{poly}(\log (NM))\log(M) \kappa^2 \epsilon^{-3})$.
>
> **Response to the Requested changes:**
>
> **R1:** We have included the text in the manuscript for the transition between the existing HSQ to QHSQ. We have also provided more details in the algorithm regarding the classical quantities computed on the quantum computer.
> * Page 5, First line after equation (12):
> Previous equations are referred to as the Hilbert space quadrature (HSQ). In this paper, we use a quantum computer to calculate these quantities, which we call the quantum Hilbert space quadrature method (QHSQ).  The variables $\mathbf{X}$ and $\mathbf{X_\mu}$ are computed on a classical computer before we feed them into the quantum computer.
> * Page 10, Section 5, First paragraph, 4 line:
> We start with the data matrix $\mathbf{X} \in \mathbb{R}^{N \times M}$ computed on a classical computer. This data matrix is exactly the same as used in Equations (11) and (12).
> * Page 10, After equation (35):
> The Gram-Schmidt decomposition works as a quantum analog of the SVD performed in (14).
> * Page 8, After equation (28):
> Quantum registers consist of multiple qubits that are used in quantum computers, similar to classical registers in classical computers (Nielsen & Chuang, 2011).
> * Page 11, Before equation (38):
> The eigenvalue estimation and the conditioned rotation correspond to the matrix inversion in (9)
>
> **R2:**
>  As suggested by the reviewer, we have added more explanations of the steps of our algorithm. In addition to the modifications in **R1** of the requested changes we added:
> * Page 10, Paragraph before Equation (37):
> From Figure (3), we can see that the ancilla register contains only one qubit, the second register contains  $\tau$ qubits, and the last two registers together contain a total of $\log_{2}\left(NM\right)$ qubits.

---

### Review · Reviewer_HaXP · 2024-11-18

**Summary Of Contributions:**

The paper proposes a quantum algorithm for Gaussian process quadrature problems, as a modification of a recently developed low-rank quantum Gaussian process regression algorithm.

**Audience:**

Yes

**Broader Impact Concerns:**

No.

**Claims And Evidence:**

Yes

**Requested Changes:**

- Writing: To help general ML readers understand math, it would be better to explain in more detail the binary representation of quantum states, e.g., how Eq. 17 can be converted into Eq. 20 (I can vaguely figure out the math derivatios there but still not very sure), $\ket{j_1} = \ket{0} + e^{2\pi i0.j_1} \ket{1}$ in Eq. 21.

- What does quantum register mean? Please explain.

**Strengths And Weaknesses:**

### Strengths

The paper is structure-wise. Sections 2 and 3 are quite useful for readers to get the background information.

The paper provides most of the math details, but it could be improved by adding intuitive descriptions to help readers understand (see requested changes)

### Weakness
It seems that the quantum algorithm doesn't outperform classical algorithms significantly. For example, Fig. 6 shows the QHSQ (R=4) deviates more from the mean than GPQ. This makes me doubt whether it is necessary to consider complicated quantum algorithms that need to be implemented by quantum computers that are still far from mature. I think some discussions are needed.

Also, from the Intro and Conclusion, it is not easy to distinguish the real contribution of the present study. It looks like the current study is a combination of recent quantum techniques. Is it possible the authors also compare the proposed quantum algorithm with other quantum algorithms?

Minor:
- Page 1, HHL: please provide its full name.

- Fig. 1 caption: The complexity of FFT is $\mathcal{O}(n \log n)$, while the caption says it is $\mathcal{O}(n2^n)$. Did I misunderstand something here?

---

> ### Author Response · Authors · 2024-11-29
> **Response reviewer HaXP 1/2**
>
> We are thankful to the reviewer for pointing out the contribution of our paper.  Also, We would like to thank you for your insightful suggestions and comments. We have revised our manuscript by incorporating your suggestions and comments. Below is our response to your comments.
>
> **Response to Weaknesses:**
>
> **R1:** Our quantum algorithm does not outperform the classical algorithm in terms of accuracy. Our algorithm solves the same classical problem in less time.  The purpose of the simulations is to demonstrate that our algorithm works well in practice and can be used for complex problems when fault-tolerant quantum computers become available. Our algorithm is more suitable for fault-tolerant quantum computers. We aim to provide ready application solutions when fault-tolerant quantum computers are available. The current QHSQ simulation results deviate from the GPQ due to the limited resources available to fully simulate the dynamics of the quantum algorithm. The accuracy can be improved by using computers that have the capability to represent more qubits' information. We have included the discussion in the manuscript for this.
> * Page 14 and Page 15, Conclusion section, line 5:
> We have also demonstrated the performance of the method using numerical simulations run on a classical simulator of a quantum computer, which shows that the algorithm works in practice. Our quantum algorithm computes the same solution as obtained from a classical computer but with accelerated speed. The simulation results slightly deviate from the classical GPQ because we use an approximate implementation of the unitary operation for qPCA along with a limited number of qubits.
>
> The proposed quantum algorithm is suitable for a fault-tolerant quantum computer with enough quantum registers to implement the algorithm precisely.
>
> **R2:** We have now outlined the contribution of the paper more precisely. The algorithm is indeed based on existing quantum computing components (QFT, phase estimation, QPCA) in the same sense that classic GPR/GPQ is based on existing methods for matrix multiplication and inversion.
> The paper develops a quantum Bayesian quadrature algorithm for numerical integration. Other Gaussian process regression methods could have been used, but we decided to go with the low-rank Bayesian quadrature rule scheme, because it is computationally efficient, and because the GPR version of it was already developed in Farooq. et all (2024).
>
> A comparison with current quantum algorithms for classical numerical integration would in principle be possible, but the other numerical integration methods are solving a different task and the comparison would be unfair.
> * Page 3, First paragraph, Line 4:
> These integration methods for classical numerical integration on a quantum computer have been proposed as an application of efficient state preparation for quantum amplitude estimation (see Martínez de Lejarza et al., 2023).
> * Page 3,  Second paragraph, Line 4:
> In this paper, we develop a low-rank quantum Gaussian process quadrature method for numerical integration employing a low-rank Gaussian process prior, using the GPR method in Farooq. et. al (2024) as the starting point.
>
> **Responses to Minor changes:**
>
> **R1:** As suggested, we have included the full name of the HHL algorithm on page 2 of the manuscript.
>
> **R2:** A quantum state uses $n$ qubits to encode $N = 2^{n}$ data points to represent information on a quantum computer.
>  By substituting $N=2^n$  in place of $n$ in FFT complexity expression, we obtain $O(N\log N)=O(n 2^{n})$,  which a quantum computer can solve with $O(n^2)$ complexity. We have revised our text to avoid confusion.
> * Page 7, Last paragraph, 4 line:
> In contrast, representing the same amount of information on a classical computer requires $N=2^{n}$ classical bits. Therefore, implementing the discrete Fourier transform using the classical fast Fourier transform (FFT) would require a computational complexity of $O(n 2^n)$.

---

> ### Author Response · Authors · 2024-11-29
> **Response reviewer HaXP 2/2**
>
> **Response to requested changes:**
>
> **R1:** We have added more details in order to help the reader understand the binary representation of quantum states.
> * Page 6,  Third line before equation (20):
> Similarly, a decimal number $0.j_{1}j_{2}\cdots j_{l}$ can be expressed in a binary representation as $j_{1}/2 + j_{2}/4 + \cdots + j_{l}/2^{l}$.
>
> * Page 6 and Page 7,  Equation (21) and afterward text:
> $$H|{j_t}\rangle=\frac{1}{\sqrt{2}}\left(|{0}\rangle+e^{\frac{2\pi \dot{\iota}}{2}j_{t}}|{1}\rangle\right) = \frac{1}{\sqrt{2}}\left(|{0}\rangle+e^{2\pi \dot{\iota} 0.j_{t}}|{1}\rangle\right),$$
>
> where we represented the $j_t/2$ as a decimal number $0.j_t$. The Equation (21) can be seen as follows: when $j_t=0$, we have $H|{0}\rangle=\frac{1}{\sqrt{2}}\left(|{0}\rangle+|{1}\rangle\right)$,  and when $j_t=1$ we have $H|{1}\rangle=\frac{1}{\sqrt{2}}\left(|{0}\rangle-|{1}\rangle\right)$.
>
> * Page 7,  First paragraph, Fifth line:
> $ CR_{k}|{1j_{t}}\rangle=e^{\frac{2\pi \dot{\iota}}{2^{k}}j_{t}}|{1j_{t}}\rangle$
>
> * Page 7,  Equation (22):
> $$\begin{align}
>     H|{j_{1}}\rangle|{j_2 \cdots j_n}\rangle=\frac{1}{2^{1/2}}\left(|{0}\rangle+e^{\frac{2\pi \dot{\iota}}{2}j_{1}} |{1}\rangle\right)|{j_2 \cdots j_n}\rangle \\\\
> =\frac{1}{2^{1/2}}\left(|{0}\rangle+e^{2\pi \dot{\iota}0.j_{1}} |{1}\rangle\right)|{j_2 \cdots j_n}\rangle.
> \end{align}$$
>
> * Page 7,  Equation (23)
> $$\begin{align}
>     CR_{2}|{j_2 j_{1}}\rangle |{j_3 \cdots j_n}\rangle=\frac{1}{2^{1/2}}\left(|{0}\rangle+e^{\frac{2\pi \dot{\iota}}{2}j_{1}}e^{\frac{2\pi \dot{\iota}}{4}j_{2}} |{1}\rangle\right) |{j_2 \cdots j_n}\rangle \\\\=\frac{1}{2^{1/2}}\left(|{0}\rangle + e^{2\pi \dot{\iota}0.j_{1}j_{2}} |{1}\rangle \right)|{j_2 \cdots j_n}\rangle.
> \end{align} $$
>
> * Page~7,  Equation (24):
> $$\begin{align}
> |{\xi_1}\rangle=\frac{1}{2^{1/2}}\left(|{0}\rangle+e^{2\pi \dot{\iota}\left(j_1/2 +j_2/4 +\cdots+j_n/2^n\right)} |{1}\rangle\right) |{j_2 \cdots j_n}\rangle\\\\
> =\frac{1}{2^{1/2}}\left(|{0}\rangle+e^{2\pi \dot{\iota}0.j_{1}j_{2}\cdots j_{n}} |{1}\rangle\right) |{j_2 \cdots j_n}\rangle,
> \end{align}$$
>
> **R2:** A quantum register stores multiple qubits, similar to classical registers which store multiple bits. We specify in our Figures 3 and 4, the number of qubits that each register contains. All of our ancilla registers contain one qubit. We have revised our text as follows:
> * Page 8, After equation  (28):
> Quantum registers consist of multiple qubits that are used in quantum computers, similar to classical registers in classical computers (Nielsen \& Chuang, 2011).
>
> * Page 10, Paragraph before Equation (37):
> From Figure (3), we can see that the ancilla register contains only one qubit, the second register contains  $\tau$ qubits, and the last two registers together contain a total of $\log_{2}\left(NM\right)$ qubits.

---

> > ### Comment · Reviewer_HaXP · 2025-01-16
> >
> > Thanks for the authors' revision which have addressed all my suggested revisions. I recommend it to a publication.

---

### Decision · Action_Editor_HbSm · 2025-02-16

**Recommendation:** Accept as is

**Comment:**

The reviewers were unanimous in recommending acceptance of the paper. The reviewers commended the paper for being well-written and illustrated. It provides ample background material in order to improve accessibility for members of the community unfamiliar with quantum algorithms. The authors diligently addressed the questions of the reviewers and significantly improved readability during the revision process. One reviewer commented that the paper is "solid" and a "serious piece of work and as such it merits to be accepted".

**Audience:**

There is a precedent for quantum algorithms in TMLR (e.g. Wiedemann et al. (2023)), so the audience criterion is satisfied.

**Claims And Evidence:**

This paper provides a novel implementation of Gaussian process quadrature (GPQ) on a quantum computer. The implementation itself is presented in Section 5, and the numerical simulations confirming the analytic properties are contained in Section 6.